Technical Report

# Locityper enables targeted genotyping of complex polymorphic genes

**Timofey Prodanov** [1,2] ✉, **Elizabeth G. Plender**[3,4], **Guiscard Seebohm** [5], **Sven G. Meuth**[6], **Evan E. Eichler** [3,7] **& Tobias Marschall** [1,2] ✉

The human genome contains many structurally variable polymorphic loci, including several hundred disease-associated genes, almost inaccessible for accurate variant calling. Here we present Locityper, a tool capable of genotyping such challenging genes using short-read and long-read whole-genome sequencing. For each target, Locityper recruits and aligns reads to locus haplotypes, for instance, extracted from a pangenome, and finds the likeliest haplotype pair by optimizing read alignment, insert size and read depth profiles. Across 256 challenging medically relevant loci, Locityper achieves a median quality value (QV) above 35 from both long-read and short-read data, outperforming state-of-the-art Illumina and PacBio HiFi variant calling pipelines by 10.9 and 1.7 points, respectively. Furthermore, Locityper provides access to hyperpolymorphic *HLA* genes and other gene families, including *KIR*, *MUC* and *FCGR*. With its low running time of 1 h 35 m per sample at eight threads, Locityper is scalable to biobank-sized cohorts, enabling association studies for previously intractable disease-relevant genes.

Single-nucleotide variants (SNVs) are the most abundant class of genetic variants segregating in the human population and are at the same time easy to access using microarray or short-read sequencing platforms. Unsurprisingly, virtually all genome-wide association studies (GWAS) seeking to map genotypes to phenotypes have been focusing on SNVs. In contrast, structural variants (SVs), which are 50 bp in size or longer, are much more challenging to characterize; more than half of all SVs per sample are missed by short-read-based variant discovery[1–3], despite their biomedical relevance[4,5]. Almost 750 genes contain 'dark' protein-coding exons, where read mapping and variant calling cannot be adequately performed[6]; around 400 medically relevant genes are almost inaccessible because of their repetitive nature and high polymorphic complexity[7]. Of them, 273 genes are widely used for variant calling and assembly benchmarking[8,9]. Long-read technologies are needed to address this problem[10–12] and recent long-read-based

genome assembly strategies indeed led to haplotype-resolved genome assemblies of diploid samples that routinely resolve many previously intractable complex genetic loci[13,14]. Nevertheless, long-read sequencing of large cohorts remains prohibitively expensive, signifying the need for accurate short-read-based genotyping.

In the meantime, high-quality assemblies are available for hundreds of human haplotypes and give rise to a pangenome reference[2,8,15]. The genetic variation encoded therein can serve as a basis for genotyping workflows by mapping reads to a pangenome graph[16,17] or through *k*-mer-based genome inference[18]. While genome inference with Pangenie[18] has expanded the set of accessible SVs considerably[8], it exhibits limitations at complex loci with few unique *k*-mers. As an alternative strategy, methods for targeted genotyping of genes of special interest, such as the *HLA*, *KIR* and *CYP2* gene families, have been developed[19–24].

[1]Institute for Medical Biometry and Bioinformatics, Medical Faculty, Heinrich Heine University, Düsseldorf, Germany. [2]Center for Digital Medicine, Heinrich Heine University, Düsseldorf, Germany. [3]Department of Genome Sciences, University of Washington School of Medicine, Seattle, WA, USA. [4]Basic Sciences Division and Computational Biology Program, Fred Hutchinson Cancer Center, Seattle, WA, USA. [5]Institute for Genetics of Heart Diseases, Department of Cardiovascular Medicine, University Hospital Münster, Münster, Germany. [6]Department of Neurology, Medical Faculty, Heinrich Heine University, Düsseldorf, Germany. [7]Howard Hughes Medical Institute, University of Washington, Seattle, WA, USA. ✉e-mail: timofey.prodanov@hhu.de; tobias.marschall@hhu.de

In this study, we propose a new tool, called Locityper, to leverage genome assemblies in a pangenome reference or custom collection of locus alleles for fast targeted genotyping of complex loci. Locityper is a general-purpose genotyper that can efficiently process both short-read and long-read data; it integrates a range of different signals based on read depth, alignment identity and paired-end distance in a statistical model to infer genotype likelihoods. This provides an opportunity to genotype and analyze a diverse set of previously understudied genes for already available large sequencing datasets, such as the 1000 Genomes Project cohort and large biobanks like the All-of-Us[25] program and the UK Biobank (UKB)[26], where disease association studies can be performed.

## Results

### Overview of the method

Locityper is a targeted genotyping tool designed for structurally variable polymorphic loci. For every target region, Locityper finds a pair of haplotypes (locus genotype) that explain the input whole-genome sequencing (WGS) dataset in a most probable way. Locus genotyping depends solely on the reference panel of haplotypes, which can be automatically extracted from a variant call set representing a pangenome, or provided as an input set of sequences. Before genotyping, Locityper efficiently preprocesses the WGS dataset and probabilistically describes read depth, insert size and sequencing error profiles. Next, Locityper uses minimizers to recruit reads to all target loci simultaneously.

At each locus, Locityper estimates a likelihood for every possible locus genotype by distributing recruited reads across possible alignment locations at the corresponding haplotypes (Fig. 1). The likelihood function is defined in such a way to prioritize read assignments with a smaller number of sequencing errors; plausible insert sizes across the read pairs; and stable read depth without excessive dips or rises. We show that finding a maximum likelihood read assignment can be formulated as an integer linear programming (ILP) problem or identified through stochastic optimization (Methods). Finally, Locityper identifies a genotype with the highest joint likelihood and outputs the most probable read alignments to the two corresponding haplotypes.

### Locityper accurately genotypes challenging loci

To evaluate Locityper's targeted genotyping accuracy, we used a reference panel of 90 haplotypes from phased whole-genome assemblies[8] across 256 target loci (Methods) covering 13.9 Mb and fully encompassing 265 challenging medically relevant (CMR) genes[7] and 23 other protein-coding genes (Supplementary Table 1).

To measure the haplotyping error, we calculated sequence divergence between actual and predicted haplotypes (Fig. 2a) and corresponding Phred-like[27] quality values (QVs), which are widely used for genome assembly evaluation[28]. Then, we distributed haplotype predictions into five bins based on their QV (<17, 17–23, 23–33, 33–43 and ≥43), where a haplotype from the last two bins (QV ≥ 33) differs from an actual haplotype by no more than 5 bp per 10 kb (Fig. 2b), which is competitive with long-read genome assemblies from Oxford Nanopore Technologies (ONT) data[29]. Note that the haplotypes were compared across the whole locus, including both coding and noncoding regions, which avoids the need for gene annotations on highly variable haplotypes.

First, we genotyped 40 Illumina WGS datasets from the Human Pangenome Reference Consortium (HPRC) cohort. Each dataset was processed using the leave-one-out (LOO) approach, where the two relevant sample haplotypes were excluded from the reference panel. Across 20,350 cases where locus haplotypes were fully assembled, Locityper achieved a median QV = 35.27, with 58.8% haplotypes having QV ≥ 33 (15.2% for QV ≥ 43). On the other hand, 9.1% haplotypes had QV = 17–23 and 5.1% haplotypes had QV ≤ 17 (Figs. 2c and 3). Instead of unmapped reads, Locityper can process existing alignments,

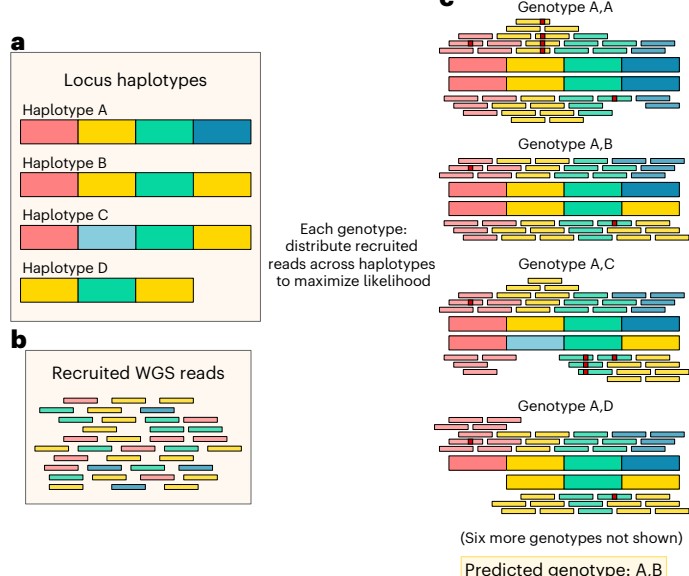

**c**
Genotype A,A

Genotype A,B

Genotype A,C

Each genotype: distribute recruited reads across haplotypes to maximize likelihood

Genotype A,D

(Six more genotypes not shown)

Predicted genotype: A,B

**a**
Locus haplotypes

Haplotype A

Haplotype B

Haplotype C

Haplotype D

**b**
Recruited WGS reads

**Fig. 1 | Illustration of the locus genotyping approach. a**, Reference panel of four locus haplotypes *(A–D)*. **b**, WGS reads, recruited to any of the haplotypes. For illustrative purposes, haplotypes and reads are colored using homologous blocks (information, unavailable to Locityper). **c**, Optimal assignments of reads to various genotypes, where the small red squares show read alignment mismatches or indels. Genotype *A,B* has the highest joint likelihood because of a small number of alignment errors and no lack or excess of read depth.

substantially accelerating the read recruitment stage. This does not lead to lower accuracy; Locityper predictions for ten mapped WGS datasets showed virtually identical results (median QV = 35.25; Supplementary Table 2).

Even though HPRC assemblies are very accurate, they may include assembly or phasing errors, especially at challenging loci. To remove this factor from the performance analysis, we used ART Illumina[30] to simulate 44 short-read datasets and processed them with Locityper. As expected, the tool showed higher accuracy on simulated datasets, producing a median QV = 35.65, with 60.7% and 4.0% haplotypes receiving QV ≥ 33 and <17, respectively (Supplementary Fig. 1a).

Locityper is not limited to short reads and can process various long-read WGS datasets, including PacBio HiFi and ONT data. For these technologies, Locityper achieved higher median QVs of 36.90 and 35.95, respectively, and produced 66.6% and 64.5% haplotypes with QV ≥ 33 (18.7% and 14.4% with QV ≥ 43), while only 2.9% and 2.0% haplotypes had QV < 17 (Extended Data Fig. 1 and Supplementary Fig. 1b).

**Locityper achieves near-optimal LOO accuracy.** By design, Locityper always associates an input WGS sample with two existing locus haplotypes. Therefore, Locityper LOO accuracy is limited to haplotype availability, that is, similarity between the actual haplotypes and the closest haplotype remaining in the LOO panel. Overall, 66.8% haplotypes had close counterparts in the LOO panel (QV ≥ 33; 20.0% for QV ≥ 43) (Fig. 2d and Extended Data Fig. 2). Inversely, 1.2% and an additional 6.5% haplotypes were dissimilar from any unrelated haplotype (QV < 17 and 17–23).

An optimal solver, which always finds the closest genotype from the LOO panel, would achieve a median QV = 36.93, just 1.66 points higher than Illumina-based Locityper and 0.03 higher than HiFi-based. For Illumina datasets, Locityper underperforms on average by just 2.03 QV points compared to the theoretical best, with 95.1% (86.8%) haplotypes trailing by under ten (five) QV points (Fig. 2h). Even further, across PacBio HiFi datasets, Locityper predictions differ from optimal by 0.72 QV points on average; this number drops down to 0.54 when considering well-represented haplotypes (availability ≥33).

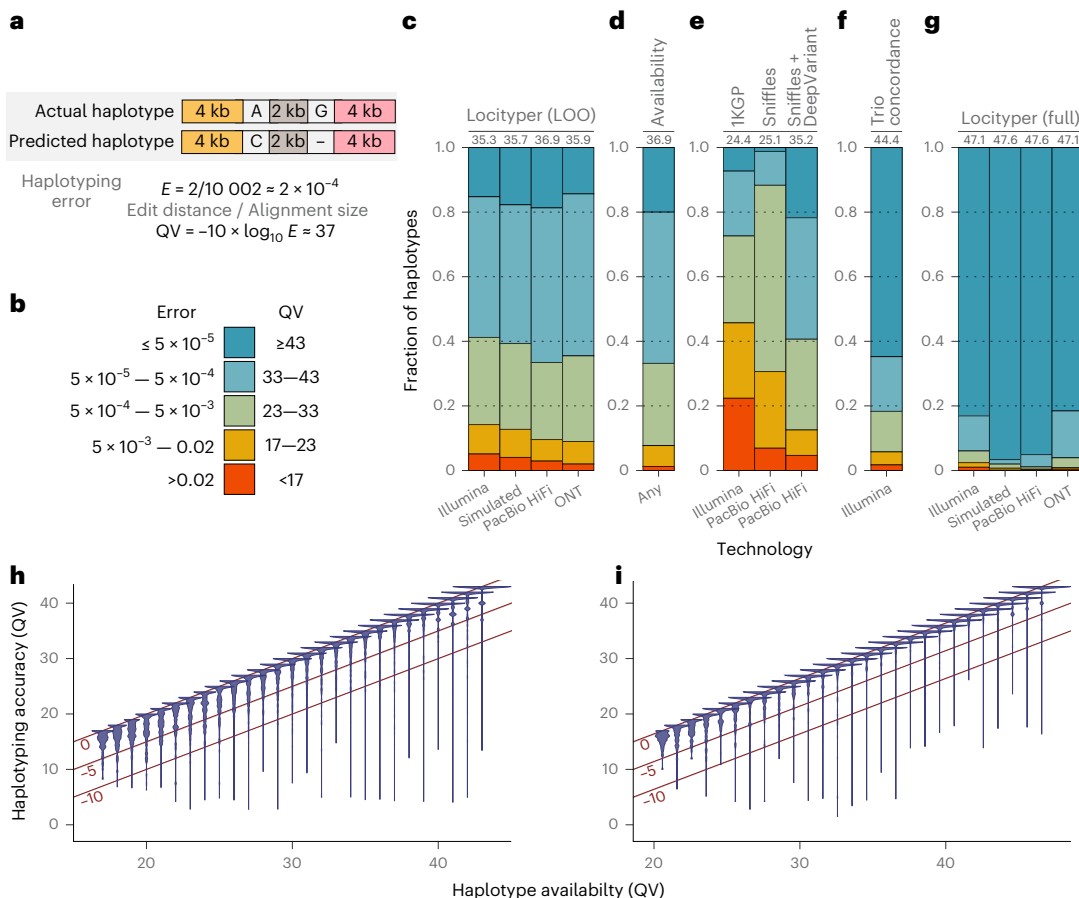

**Fig. 2 | Haplotype accuracy definition and analysis at 256 CMR loci. a**, The haplotyping error was calculated as the sequence divergence between actual and predicted haplotypes. The QV is a Phred-like transformation of the haplotyping error. **b**, Approximate correspondence between haplotyping error and QV bins. **c**–**g**, Fraction of haplotypes across 256 loci and multiple samples, distributed into five QV bins (Supplementary Table 2). The median QV is shown above each bar. **c**, Locityper accuracy in the LOO configuration. **d**, Haplotype availability

(QV between actual and closest available LOO haplotypes). **e**, Haplotyping accuracy of the 1KGP call set, as well as Sniffles and Sniffles + DeepVariant variant calling. **f**, Concordance of Locityper predictions across 563 unrelated trios. **g**, Locityper accuracy using the full reference panel. **h,i**, Correspondence between haplotype availability and haplotyping accuracy based on Illumina (**h**) and PacBio HiFi (**i**) WGS datasets. The red lines mark a 0, 5-point and 10-point QV loss.

Overall, 98.7% (96.0%) HiFi-based haplotypes were within the ten (five) point margin (Fig. 2i).

This analysis shows that Locityper performs extremely well when required haplotypes are present in the reference panel, and achieves near-optimal accuracy with only limited haplotype sets. Growing numbers of haplotypes in pangenomes[15] are likely to increase Locityper accuracy even further.

**Locityper outperforms variant calling pipelines.** By identifying the two most similar locus haplotypes to a given WGS dataset, Locityper effectively infers the two haplotype sequences at a locus. This provides an opportunity to benchmark Locityper against any phased variant call set, which likewise can be interpreted as a prediction of both haplotype sequences. Consequently, we evaluated the New York Genome Center (NYGC) call set for the expanded 1KGP (1000 Genomes Project) cohort of 3,202 samples[3], of which 39 have HPRC assemblies. Even though the NYGC pipeline uses state-of-the-art variant callers, 1KGP haplotypes had significant divergence from the actual sample haplotypes: only 27.4% haplotypes achieved QV ≥ 33 and another 22.3% haplotypes had QV < 17, while the median QV was 24.41, almost 11 points smaller than Locityper on Illumina reads (Fig. 2e and Extended Data Fig. 3).

While short-read datasets are difficult to genotype at complex loci, PacBio HiFi data are arguably the easiest. To put Locityper performance in perspective we examined phased SV calls, generated by Sniffles[31] for

20 HiFi datasets. As Extended Data Fig. 4a shows, Sniffles alone did not achieve high levels of accuracy, producing a median QV = 25.09. Combining SVs with short variant calls, produced by DeepVariant[32], raised the median QV to 35.19, which is 1.71 points behind Locityper on the same data and 0.08 points behind Illumina-based Locityper. While Sniffles + DeepVariant (Extended Data Fig. 4b) produced a larger fraction of poor haplotypes (4.7% and 7.9% with QV < 17 and 17–23 against 2.9% and 6.7% for Locityper), this pipeline also produced a bigger share of extremely accurate haplotypes (21.7% against 18.7%), probably because of Locityper's inability to call new variants.

**Locityper produces concordant trio predictions.** Additionally, we genotyped the full 1KGP cohort of 3,202 Illumina WGS samples, including 563 trios independent from the HPRC cohort. At each of the target loci and for each trio we calculated concordance, that is, the similarity between child and parent haplotypes (Methods). As Fig. 2f shows, the vast majority of trio haplotypes were concordant: 64.8% and 81.7% with QV ≥ 43 and ≥33, respectively. Moreover, the median concordance QV surpassed 44.4 and was over 43 at 90% of the loci (Extended Data Fig. 5).

**Almost perfect accuracy with a full reference panel.** Finally, we examined Locityper's ability to accurately identify true sample haplotypes using a full reference panel. This experiment should mimic future pangenomes, where almost all haplotypes present in the population

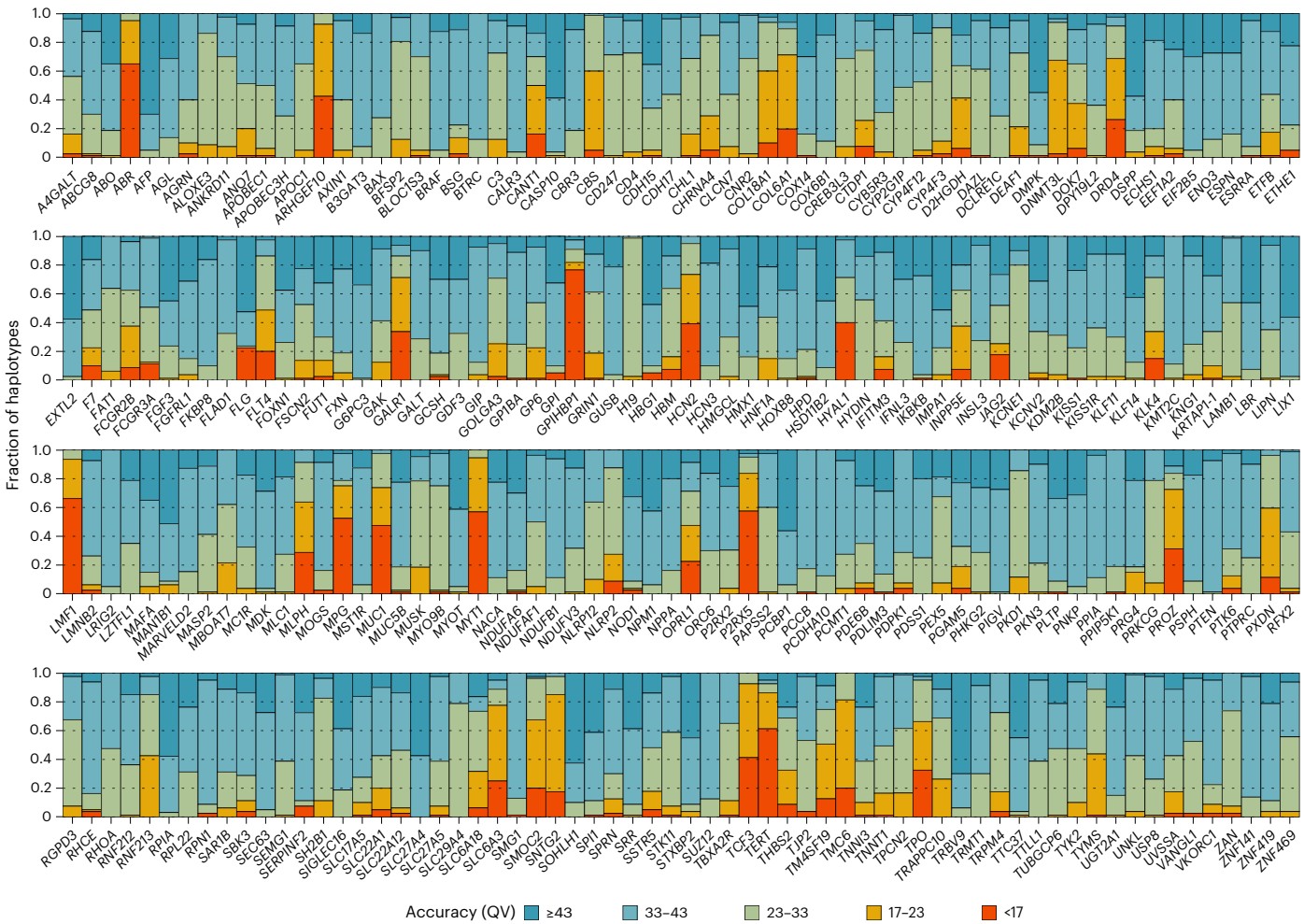

**Fig. 3 | Locityper haplotyping accuracy using an LOO reference panel for 40 Illumina WGS datasets.** Predicted haplotypes across 256 CMR loci were binned into five groups according to their haplotyping QV.

would also exist in the reference panel. At each of the sequencing technologies, Locityper achieved an extremely high median QV (>47) and produced more than 93% haplotypes with QV ≥ 33. Illumina-based and ONT-based haplotypes showed slightly lower accuracy: 83.1% and 81.6% had QV ≥ 43, respectively, while only 1.0% and 0.4% had QV < 17. On the other hand, simulated short reads and PacBio HiFi datasets produced almost perfect haplotypes: 96.6% and 95.1% with QV ≥ 43 and ≈0.1% with QV < 17 (Fig. 2g, Extended Data Fig. 6 and Supplementary Fig. 2). A variant call set obtained from the Locityper haplotypes using the full reference panel and Illumina data showed a significantly higher $F_1$ score than the 1KGP call set, as well as higher precision and recall compared to the pangenome-based variant caller Pangenie[18] (Supplementary Fig. 3 and Supplementary Information).

### Locityper accurately genotypes *HLA* and *KIR* genes
To evaluate Locityper's ability to genotype hyperpolymorphic genes, we examined genes from two medically relevant genomic regions: the major histocompatibility complex (*MHC*), covering over 4 Mb and over 200 genes[33], and the *KIR* gene cluster spanning 150 kb and 17 genes[34]. The two regions contain extremely polymorphic *HLA* and *KIR* genes, which have an essential role in adaptive and innate immune systems[35,36]. As Locityper genotypes target loci based solely on the sequences of available haplotypes, it is not limited to gene bodies and can use the intergenic sequence, gene order and presence and absence of copy-number-variable genes. As such, Locityper can predict missing genes by selecting padded haplotypes that lack the gene of interest.

Multiple specialized tools have been developed for genotyping the *MHC* locus[19,22,37], the newest being T1K[23], a state-of-the-art[38] genotyper for *HLA* and *KIR* genes that is capable of processing whole-genome and whole-exome short-read sequencing data. To compare T1K and Locityper accuracy, we genotyped 40 Illumina HPRC WGS datasets at 26 genes and 14 pseudogenes from the *MHC* locus and 14 genes and three pseudogenes from the *KIR* locus, all combined into 33 target loci with a sum length of 1.15 Mb.

In the LOO configuration, at the *MHC* locus, Locityper achieved a full match with assembly-based allele annotation (correctly predicted all fields in the *HLA* nomenclature[39,40]) in 88.8% cases, compared to T1K's 64.1% (Fig. 4a). At the same time, the two methods correctly predicted the protein product (second nomenclature field) in 95.1% and 78.2% of cases, respectively. Meanwhile, at the *KIR* gene cluster, Locityper and T1K correctly predicted protein products in 84.9% and 67.1% cases and achieved full match in 80.8% and 57.9% cases, respectively (Fig. 4b). When using the full reference panel, which also containing the input samples, Locityper achieved almost perfect accuracy: full match in 99.4% and 99.9% of cases at the *MHC* and *KIR* loci, respectively.

Unlike T1K, Locityper does not distinguish between exons, introns and intergenic space. This may result in lower accuracy when a haplotype carrying a false gene allele better explains input reads within a noncoding sequence. To handle such cases, users may use a weighted Locityper mode, giving lower weight to read depth and read alignments occurring outside exons. Using a weight of 0.1 for introns and 0.005 for intergenic regions, Locityper's accuracy rose to 96.5%

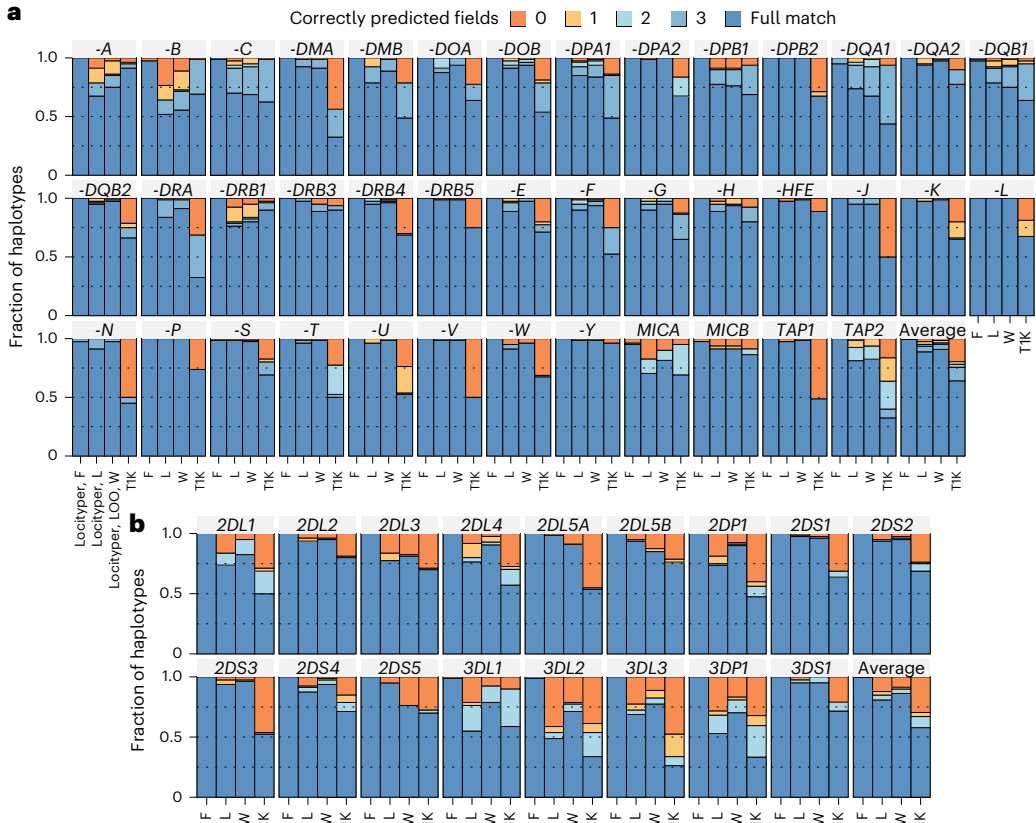

**Fig. 4 | Haplotyping accuracy for 40 HPRC samples at the *MHC* and *KIR* loci.** **a**,**b**, Subpanels showing the fraction of haplotypes, predicted with varying accuracy at 40 pseudogenes from the *MHC* locus (**a**) and 17 pseudogenes from the *KIR* gene cluster (**b**). Fully predicted alleles and correctly identified missing copies, are colored dark blue (full match) because of the different number of allele fields in the *HLA/KIR* gene nomenclature[39,58]. Otherwise, haplotypes are colored according to the number of correctly predicted fields. Accuracy is shown for Locityper with the full reference panel (F), Locityper in the LOO setting with and without weights (denoted L and W, respectively) and T1K. The last entry in each panel shows the average accuracy across all corresponding genes and pseudogenes.

(protein product) and 90.8% (full match) at the *MHC* locus, and to 89.9% and 86.2% at the *KIR* cluster (Fig. 4).

Some protein products were present in only one HPRC sample; consequently, such samples cannot be correctly annotated by Locityper in the LOO setting. Such cases explained 64.6% and 36.2% of all errors made by Locityper in the weighted mode at the *MHC* and *KIR* loci, respectively (Extended Data Fig. 7). This is especially noticeable at the hyperpolymorphic *HLA-A*, *HLA-B* and *HLA-DRB1* genes, where protein groups were missing from the LOO panel in 10–22% of cases, which explains the vast majority of Locityper errors. At the same time, T1K often predicted a smaller copy number than required, explaining 79.1% and 24.6% of all errors at the *MHC* and *KIR* clusters, respectively. When ignoring these two error types (missing copy and unavailable protein groups), Locityper notably outperformed T1K in predicting protein products: 99.0% against 94.5% at the *MHC* locus, and 94.6% against 73.0% at the *KIR* gene cluster. Overall, the general-purpose tool Locityper performed in a competitive manner even when compared to T1K, which was specifically designed for *HLA* and *KIR* genes. However, accurate genotyping of the most diverse genes would still probably benefit from larger pangenome sizes.

## Accurate genotyping of disease-relevant gene families

Although the set of CMR genes included a wide variety of genetically diverse genes, several important polymorphic gene families were underrepresented in it. The mucin genes are a highly heterogeneous gene family (*MUC1–MUC24*)[41]. Mucin genes encode large glycoproteins that are essential to barrier maintenance and the defense of epithelial tissues. All canonical mucins harbor a large exon that contains variable number tandem repeats (VNTRs), whose sequences vary per mucin, yet each extensively encode serine and threonine residues for glycosylation[42]. The gene family can be broken up into two subgroups: tethered and secreted mucins. In tethered mucins, single VNTR domains contain variation in total motif copy number and motif usage (Fig. 5a). Secreted mucins harbor potential variation in VNTR domain copy number, VNTR motif copy number, VNTR motif usage and cysteine domain copy number[43,44] (Fig. 5b). The presence of these repetitive sequences makes mucins both highly polymorphic and difficult to accurately sequence and genotype using short reads.

Locityper leverages information about both read depth and read alignment for genotyping; therefore, the tool is well suited to characterizing mucin genetic variation. Based on 39 HPRC Illumina WGS datasets, Locityper (LOO) haplotypes achieved on average a 10.5 higher QV compared to the 1KGP call set across 15 examined *MUC* loci, with the largest improvement observed at *MUC6* and *MUC16* with 29.7 and 18.5 higher QV, respectively (Fig. 5c). The only negative QV difference between Locityper and 1KGP was observed at the non-gel-forming *MUC7* gene, where the two haplotype sets showed very high QV values (43.5 and 44.2, respectively).

Further examples of genes that are challenging to address with standard calling techniques are *FCGR2B* and *FCGR3A*, encoding receptors for the Fc region of the IgG complexes[45,46]. IgG binding to FCGR2B induces the immune complexes of phagocytosis and endocytosis and thus establishes the basis of antibody production by B cells. The second receptor, FCGR3A, is expressed on natural killer cells as an integral

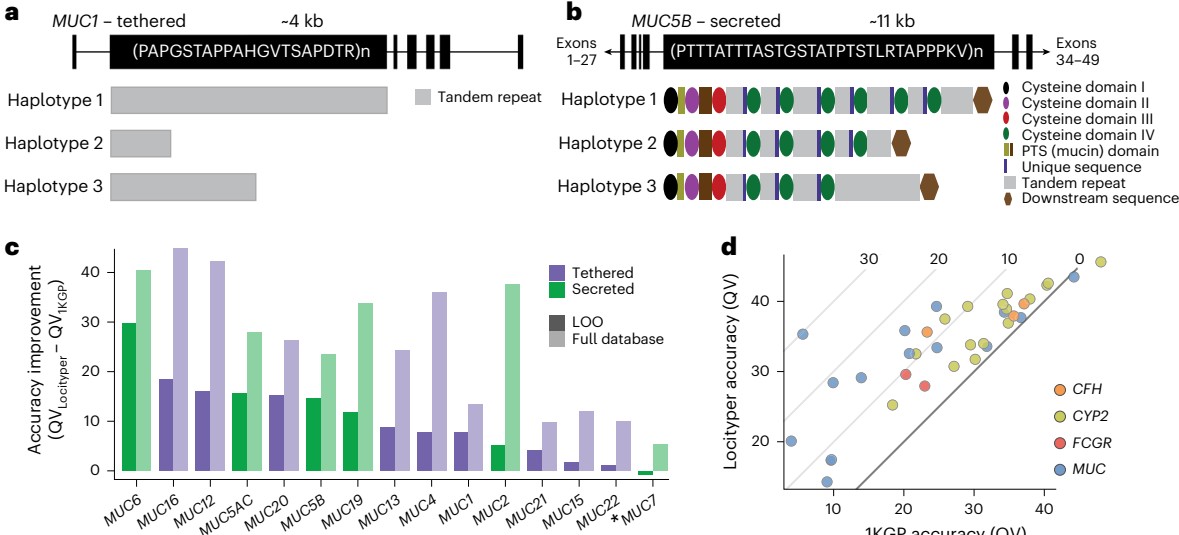

**Fig. 5 | Lociotyper can accurately genotype mucin and other gene families.**
**a**, Gene model of *MUC1*, a mucin tethered to the surface of epithelial cells. *MUC1* harbors a 20-amino-acid VNTR repeat sequence and is highly polymorphic in VNTR length[59], as represented by the example haplotypes 1–3. **b**, Gene model of *MUC5B*, a secreted, gel-forming mucin that is important for homeostasis in the lungs. *MUC5B* encodes an irregular 29-amino-acid VNTR motif that is broken up into separate VNTR domains by cysteine domains. The number of VNTR domains, cysteine domains and VNTR motifs could each contribute to polymorphism among haplotypes at this locus[60]. **c**, Difference in average haplotyping accuracy (QV) between Lociotyper and the 1KGP call set at 15 mucin genes based on 39 Illumina WGS datasets. Improvement for the LOO setting and the full Lociotyper database are shown using dark and light shades, respectively. Tethered and secreted mucins are shown in purple and green; the only non-gel-forming secreted mucin *MUC7* is marked with an asterisk. **d**, Lociotyper (LOO) and 1KGP call set average genotyping accuracy (QV) across four gene families: *CFH* (orange); *CYP2* (light green); *FCGR* (red); and *MUC* (blue). The diagonal black line shows the zero improvement boundary and the diagonal gray lines show a QV improvement of 10, 20 and 30. PTS, proline (P), threonine (T), serine (S).

membrane glycoprotein[46] and has a central role in limiting viral load and viral propagation in a memory-like manner[47]. Genetic variations in both genes have been associated with systemic lupus erythematosus[48] and other immune disorders[49]. However, genetic analyses of the *FCGR* genes using high-resolution short reads have been notoriously difficult because of recent gene duplication and diversification processes[50]. Nevertheless, at the *FCGR2B* and *FCGR3A* receptor genes, Lociotyper (LOO) improves the average QV by 4.95 and 9.3 points, respectively, compared to the 1KGP call set (23.0 to 27.9 and 20.3 to 29.6) (Fig. 5d). A larger reference panel would probably improve Lociotyper's ability to genotype *FCGR* genes even further because the tool achieves much higher accuracy (35.6 and 54.0) when using its full reference panel.

Moreover, Lociotyper (LOO) achieves significant QV improvement (12.3) at the *CFH* gene, which is associated with age-related vision loss and kidney disorders[51,52]. Finally, Lociotyper showed on average a 4.6 higher QV across 16 protein-coding *CYP2* genes that have a major role in drug metabolism[53,54]. Out of the *CYP2* genes, Lociotyper achieved the highest improvement at *CYP2U1* (10.2), *CYP2A13* (10.8) and *CYP2W1* (11.6) (Fig. 5d).

### Runtime and memory usage

Lociotyper WGS preprocessing (executed once per dataset) took on average 16 min using eight threads and consumed 15 Gb of RAM for 30× Illumina WGS datasets. If a dataset with a similar library preparation was previously processed, read mapping can be skipped, which speeds up WGS preprocessing to under 3 min. The next step, read recruitment, can simultaneously identify reads for multiple target loci. Because reading and decompressing input data was the most time-consuming operation, recruitment speed did not depend on the number of loci (1–256 tested) and lasted under 15 min on average.

Next, mapping reads to the reference panels across 256 target loci took under 19 min using eight threads; locus genotyping consumed another 45 min. Together, these two steps required approximately 15 s per target locus and 7 Gb of RAM. Lociotyper uses stringent haplotype filtering as the first genotyping step, allowing it to avoid quadratic

runtime. Thus, full analysis based on five-times-larger reference panels (obtained by artificially mutating existing haplotypes) required only three times as much time (Extended Data Fig. 8).

Altogether, Lociotyper analysis of the *MHC* and *KIR* loci, including preprocessing, required 35 min using eight threads. However, genotyping in the weighted mode was more computationally intensive, raising the total runtime to 1 h and 5 min. At the same time, T1K with eight threads required on average 2 h and 30 min and 48 min to process the *MHC* and *KIR* loci, respectively, and required 2.5 Gb of RAM. Pangenie calls variants across the whole genome; consequently, it had a heavier runtime and memory footprint: at 24 threads, its pangenome indexing (executed once) and genotyping steps took 34 min and 1 h and 40 min, respectively, and consumed 60 and 37 Gb of RAM.

In addition to unmapped data, Lociotyper and T1K can efficiently use mapped reads (in BAM/CRAM format for Lociotyper and BAM format for T1K) by only recruiting reads aligned to the regions of interest or to alternative contigs, as well as unmapped reads. Additionally, by examining existing alignments, Lociotyper can preprocess WGS datasets almost immediately. Overall, this decreases T1K runtime to 45 min and 23 min for the *HLA* and *KIR* loci, respectively, and speeds up the full Lociotyper pipeline for these genes to 10 min.

## Discussion

In this study, we present Lociotyper, a targeted method for genotyping complex polymorphic genes using both short-read and long-read WGS. Lociotyper implements fast read recruitment to a collection of target loci, and uses a carefully balanced probabilistic model to calculate genotype likelihoods based on read alignment, insert size and read depth profiles. Lociotyper uses ILP or stochastic optimization to find the most likely genotype for each target locus. Lociotyper departs from the prevalent variant-centric approach, which we argue constitutes a particular limitation for highly polymorphic loci. In contrast, our approach leverages collections of known haplotype sequences, which can be extracted from a pangenome reference or directly provided by the user. By examining larger regions around genes of interest,

Locityper inherently makes use of any available information, including the intergenic sequence, gene order, SVs and copy number of short tandem repeats. Locityper is easy to install via Docker, Singularity or Conda, only requires easy-to-obtain input files, and has a small memory footprint and significantly shorter runtime than both T1K and Pangenie.

We demonstrated Locityper's accuracy through excellent agreement to both phased genome assemblies and Mendelian consistency across the 563 family trios included in the 1KGP cohort. When evaluated across a wide range of challenging disease-associated genes, Locityper produces significantly more accurate haplotype predictions compared to state-of-the-art phased variant calling pipelines on Illumina and PacBio HiFi data. Locityper's accuracy remains consistently high across several input sequencing technologies, performing well for Illumina, simulated short reads, PacBio HiFi and ONT datasets.

At present, the size of the available collections of reference haplotypes still poses a limitation: overall, 33% haplotypes did not have a good representative (QV < 33) in the LOO reference panels (Fig. 2d). Therefore, despite Locityper's ability to predict haplotypes close to the best available, the resulting accuracy is not yet ideal for all genes of interest. Significantly larger pangenomes are presently being constructed by the HPRC[15] and we are confident that these future pangenomes will lead to a significant increase in performance on out-of-sample individuals for more complex polymorphic genes. Even now, Locityper outperforms the specialized genotyper T1K across *HLA* and *KIR* genes in a LOO setting and shows improved ability to genotype other medically relevant gene families (for example, *MUC* and *FCGR*) using short-read WGS.

As part of this study, we used Locityper to process 3,202 Illumina WGS datasets from the 1KGP and make the obtained genotypes available, which provides a resource for deeper analyses of 256 challenging target loci. Additionally, publicly available Locityper-preprocessed WGS summaries will allow for faster genotyping of genes that were not a focus of this study across the 1KGP cohort. We envision that Locityper will enable the inclusion of complex loci in GWAS[55] and PheWAS[56] analyses, especially in a larger cohort, such as the All-of-Us program[25] and the UKB[26], which promises to discover many new associations and explain missing heritability. Of note, Locityper's ability to process both short and long reads might prove especially useful for the increasing production of long reads in the context of biobank-scale sequencing efforts.

For a given locus, Locityper aims to find two existing haplotypes that would explain an input WGS dataset in the best way. Consequently, it is not designed to reconstruct a new haplotype, even if it constitutes a mixture of already known haplotypes. To address this, Locityper outputs read alignments to the top predicted genotypes, which can be used later for visual analysis or variant calling. Combined with assembly polishing[57], this could improve genotyping accuracy and allow for the reconstruction of previously unobserved alleles, a strategy that we plan to explore in future research.

Currently, two loci with significant homology, for example, part of a non-tandem segmental duplication, can only be processed independently, with potentially overlapping sets of recruited reads. Locityper mitigates this problem by tracking the number of off-target *k*-mers per read and haplotype window. Nevertheless, further improvements are conceivable, such as using a shared pool of reads for related loci, like the strategy implemented by T1K[23].

In conclusion, Locityper allows for fast and accurate targeted genotyping of challenging polymorphic loci using several sequencing technologies. With the current draft pangenome containing highly accurate phased genome assemblies, Locityper routinely achieves sequence accuracies above a QV of 33, which is comparable to genome assemblies from Oxford Nanopore data[29]. As more human haplotypes are represented in pangenomes, we expect the accuracy to improve further, which will facilitate detailed analysis of previously intractable genes, leading to improved diagnostic power and new disease associations.

## Online content

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

## Methods

In this article, we present a targeted tool, Locityper, designed for genotyping complex multiallelic loci. Locityper processes WGS data produced by several different sequencing technologies, including highly accurate short and long reads (such as Illumina and PacBio HiFi data, respectively), as well as error-prone long reads, such as PacBio CLR and Oxford Nanopore data. Locityper can efficiently analyze unmapped reads stored in various formats, as well as mapped reads from sorted and indexed BAM and CRAM files.

Broadly, the method can be split into several steps: (1) preprocessing of target loci; (2) sample preprocessing, performed once for each WGS dataset; (3) read recruitment, carried out simultaneously for multiple loci; and (4) locus genotyping and generation of BAM files with alignment to the best genotypes. These steps are described in more detail in the following sections.

### Preprocessing of target loci

Locityper uses solely locus haplotype sequences and does not require any kind of additional graph structure. Locus haplotypes can be provided directly in a FASTA file. Alternatively, Locityper can automatically extract locus haplotypes from a pangenome, provided in variant calling format (VCF) (constructed, for example, using Minigraph-Cactus[61]).

When locus haplotypes are extracted from a VCF file, Locityper tries to extend the locus in such a way that both locus ends do not overlap any pangenomic variation. Additionally, the tool tries to select a position that would produce the largest number of unique canonical $k$-mers at the edges of the locus (default edge size = 500 bp). In the default configuration, locus extension is limited to 50 kb at each side, but can fail if there is a longer SV at the locus boundary. In such cases, the user can either increase the allowed extension size or set the boundaries manually.

Finally, Locityper finds off-target $k$-mer multiplicities, calculated as the difference between canonical $k$-mer counts across the full reference genome (calculated using Jellyfish[62] with recommended $k = 25$) and the corresponding $k$-mer counts at the reference locus sequence.

### WGS dataset preprocessing

Locityper aims to probabilistically describe three features of a given WGS dataset, that is, insert size, error profile and read depth, by examining read alignments to a predefined background region. For human WGS data, we used a 4.5 Mb interval on chromosome 17q25.1 as the default background region because it contains almost no segmental duplications or other types of structural variations. Locityper first recruits input reads to the background region (see 'Read recruitment'), optionally subsamples them and then maps them to the reference genome using Strobealign[63] (short reads) or Minimap2 (ref. [64]) (long reads).

**Insert size.** Manual examination of several paired-end WGS datasets from the HPRC project[15] indicated that the negative binomial (NB) distribution fits insert size distribution the best (Extended Data Fig. 9). For a given WGS dataset, we used all fully mapped read pairs (clipping less than 2% of the read length, by default) with high mapping quality (≥20). We removed outliers by defining the maximum allowed insert size as three times the 99th percentile of the observed insert sizes, and discarded violating read pairs. Finally, we obtained the NB distribution parameters using the method of moments. During the next two preprocessing steps, we only used read pairs with insert sizes within the 99.9% confidence interval of the corresponding NB distribution.

**Error profile.** We used two distributions to describe the WGS error profiles. First, we used the beta binomial (BB) distribution to evaluate the edit distance based on read length. The distribution was fitted using the maximum likelihood estimation based on the remaining read pairs.

The obtained BB distribution was used to distinguish between true and off-target alignments at the genotyping stage.

Second, we calculated match, mismatch, insertion and deletion rates ($P_M, P_X, P_I, P_D$, respectively) and defined the alignment likelihood as the product of the corresponding rates to the power of the number of operations. For example, alignment with 100 matches, one mismatch and two insertions would have a likelihood of $P_M^{100} \times P_X^1 \times P_I^2 \times P_D^0$. Note that the probabilities do not sum up to one and are incomparable between reads of different lengths. Nevertheless, this formulation produces fast-to-calculate probabilities and provides a way to numerically compare different alignments of the same read.

**Read depth.** We split the background region into windows of fixed size based on the mean read length and assigned reads to windows based on the middle of the corresponding read alignments. Next, we counted the number of primary read alignments assigned to each window. (Only first mates were counted to preserve window independence.)[65]

For each window, we calculated the guanine-cytosine (GC) content and the fraction of unique $k$-mers in an area centered around the window. Next, we selected windows with many unique $k$-mers (≥90%) and estimated the mean read depth and variance across various GC content values using local polynomial regression[66]. NB parameters were then estimated separately for each GC content based on the smoothed mean, variance and subsampling rate (Supplementary Information).

### Read recruitment

After dataset preprocessing, Locityper recruits reads to all target loci. For that, we collected minimizers[67] from each locus and each haplotype (default: (10,15)-minimizers). Uninformative minimizers, which appear five or more times off-target, were ignored. Locityper compares read and target minimizers in parallel and recruits reads to one or several loci according to one of the following rules: short reads are recruited if a sufficient fraction of minimizers matches the target for all read ends (default: 0.7 and 0.5 for single-end and paired-end reads). Lower match fraction values lead to an increased number of unnecessarily recruited reads, which increases read mapping runtime but does not significantly affect genotyping accuracy because false positive reads are discarded at a later stage.

Only a small part of a long read may overlap a given target locus. Consequently, we recruited a long read if it contained a subregion with sufficiently many minimizer matches. For that, we used the following heuristic: matching and mismatching informative minimizers are assigned $s_+/s_-$ scores (default: +3/−1); a read was recruited if it had a continuous subsequence, with a sum score greater or equal to

$$\left\lceil 2L \frac{M(s_+ - s_-) + s_-}{m_w + 1} \right\rceil \quad (1)$$

where $L$ is the subregion length (default: 2,000 bp), $M$ is the match fraction (default: 0.5) and $2L/(m_w + 1)$ is the expected number of $(m_w, m_k)$ minimizers per $L$ base-pair sequence[68]. This heuristic is useful because it can be quickly evaluated using Kadane's algorithm[69] and is not too restrictive: shorter read subregions with a higher match rate may produce a hit, and vice versa.

### Genotype likelihood

**Read location probabilities.** After read recruitment, every target locus was genotyped independently from other loci. Reads, recruited to the locus, were aligned to all haplotypes $H$ using either Strobealign[63] or Minimap2 (ref. [64]), depending on the read type. The obtained read alignments were assigned BB $P$ values according to their edit distances and read lengths. A read pair was retained if both read ends had at least one good alignment ($P \geq 0.01$) to at least one of the haplotypes (approximately 3% read pairs discarded per locus). All alignments with BB $P < 0.001$ were discarded.

Without loss of generality, we describe the following steps for paired-end reads and use notation $\mathbf{r} = (r_1, r_2)$ to describe a read pair. Each locus haplotype $h \in H$ was split into nonoverlapping windows $W^{(h)}$ of fixed size (same size as in read depth preprocessing); furthermore, we expanded $W^{(h)}$ by adding a null window $w_\circ$. Each alignment is connected to a single window $w$ based on the middle point of the alignment, with alignment probability $P(r_j, w)$ calculated according to the precomputed error profile. Reads without proper alignment to $h$ are connected to the null window $w_\circ$; we defined $P(r_j, w_\circ)$ as $\Lambda \cdot \max_h P(r_j, h)$, that is, the probability of the best $r_j$ alignment to any haplotype, multiplied by a penalty $\Lambda$ ($10^{-5}$ by default).

The paired-end alignment probability of the read pair $\mathbf{r} = (r_1, r_2)$ to windows $\mathbf{w} = (w_1, w_2)$ can be written as $P(\mathbf{r}, \mathbf{w}) = P(r_1, w_1) \times P(r_2, w_2) \times P_{\mathrm{insert}}(\mathbf{r}, \mathbf{w})$, where the last term is calculated according to the precomputed insert size distribution. For the null windows, we defined insert size probability as the highest probability achievable under the precomputed insert size distribution. Thus, the insert size between a read end and its unmapped counterpart is assumed to be optimal to only penalize unpaired locations once. Finally, we denoted the full set of possible read pair locations on haplotype $h$ as $L^{(h)} \subset W^{(h)} \times W^{(h)}$ and defined the probability of the read pair $\mathbf{r}$ location to be $\mathbf{w}$ as the normalized alignment probability:

$$\mathcal{P}_{\mathbf{rw}} = \frac{\mathcal{P}(\mathbf{r}, \mathbf{w})}{\sum_{h' \in H} \sum_{\mathbf{u} \in L^{(h')}} \mathcal{P}(\mathbf{r}, \mathbf{u})} \qquad (2)$$

Some parts of the target loci can have high homology to other genomic regions. Consequently, we downgraded the effect of potentially misrecruited reads by setting equal probabilities to all locations for read pairs with fewer than five target-specific $k$-mers.

**Read assignment.** Without loss of generality, let us consider a diploid genotype $\mathbf{g} = (h_1, h_2)$. We combined windows across the two haplotypes $W^{(\mathbf{g})} = W^{(h_1)} \cup W^{(h_2)}$. If $h_1 = h_2$, we used two copies of each window, such that $|W^{(\mathbf{g})}|$ is always $|W^{(h_1)}| + |W^{(h_2)}|$. Similarly, we concatenated possible locations $L^{(h_1)}$ and $L^{(h_2)}$ to achieve a combined list of locations $L^{(\mathbf{g})}$.

We described read assignment to the genotype $\mathbf{g}$ using a Boolean matrix $T$, where $T_{\mathbf{rw}} = 1$ encodes the statement 'true location of the read pair $\mathbf{r}$ is $\mathbf{w}$' and every row contains exactly one true element. Probability of the read assignment $T$ given read pairs $R$ can be described as the total probability of all selected locations:

$$P(T \mid R) = \prod_{r \in R} \sum_{\mathbf{w} \in L^{(\mathbf{g})}} T_{\mathbf{rw}} \cdot \mathcal{P}_{\mathbf{rw}} \qquad (3)$$

**Read depth likelihood.** In addition to good alignment probabilities, optimal haplotypes should have stable haploid read depth. The corresponding conditional probability can be written as:

$$P(\mathrm{CN}(\mathbf{g}) = 1 \mid T) = \prod_{w \in W^{(\mathbf{g})}} P(\mathrm{CN}(w) = 1 \mid d_w(T)) \qquad (4)$$

In this equation, $d_w(T)$ denotes the window $w$ depth according to the read assignment $T$, defined as $\sum_{\mathbf{r}} \sum_u [T_{\mathbf{r},wu} + T_{\mathbf{r},uw}]$. At CN = 1, read depth follows the NB distribution with the precomputed parameters $n$ and $\psi$. Bayes' theorem with equal priors produces the following result:

$$\varphi_w(T) = P(\mathrm{CN}(w) = 1 \mid d_w(T) = d) = \frac{\mathrm{NB}(d; n, \psi)}{\sum_{c \in \{1\} \cup C_{\mathrm{alt}}} \mathrm{NB}(d; cn, \psi)} \qquad (5)$$

where alternative hypotheses are represented by a set $C_{\mathrm{alt}}$. We found it beneficial to use $C_{\mathrm{alt}} = \{0.5, 1.5\}$; in other words, a half divergence from the expected read depth was considered significant. As unmapped reads are already penalized by low alignment probabilities $P(r, w_\circ)$, we defined $P(\mathrm{CN}(w_\circ) = 1 \mid d)$ for any read depth $d$.

**Window and read weights.** Low-complexity regions, and short and long repeats, evoke difficulties in read sequencing, recruitment and alignment. To assign window weights in a continuous fashion, we defined the following two parametric function $\vartheta : [0, 1] \mapsto [0, 1]$ as:

$$\vartheta(x; \eta, q) = \begin{cases} 0 & \text{if } x = 0, \\ \dfrac{1}{\left( \frac{\eta}{x} \times \frac{1-x}{1-\eta} \right)^q + 1} & \text{otherwise} \end{cases} \qquad (6)$$

$\vartheta$ exhibits several useful properties: it is a strictly increasing smooth function such that $\vartheta(0) = 0$ and $\vartheta(1) = 1$. The location parameter $\eta \in (0, 1)$ defines the break point $\vartheta(\eta; \eta, q) = 1/2 \; \forall q$, while the power parameter $q$ controls the slope of the function, with larger $q$ producing larger derivative $\vartheta'(\eta; \eta, q)$ (Extended Data Fig. 10). Finally, we defined window $w$ weight $\zeta_w = \vartheta(x_1; \eta_1, q_1) \times \vartheta(x_2; \eta_2, q_2)$ based on the fraction of the locus-specific $k$-mers $x_1$ and linguistic sequence complexity $x_2 = U_1 U_2 U_3$, where $U_i$ is the fraction of unique $i$-mers in window $w$ of the maximal possible number of distinct $i$-mers[70], with the default parameters $\eta_1 = 0.2$, $\eta_2 = 0.5$ and $q_1 = q_2 = 4$.

Locityper accepts explicit user-defined weights for each base pair of the input haplotypes, useful, for example, for downweighting noncoding sequence. In such cases, $\zeta_w$ is multiplied by the average weight across window $w$, while each read receives its own weight based on the maximum explicit weight under the primary alignments of both read ends. After that, read weights are used as multipliers to log-location-probabilities.

**Combined likelihood and likelihood update.** Not accounting for window weights, combined likelihood for a genotype $g$ and read assignment $T$ can be calculated as:

$$\begin{aligned} P(\mathrm{CN}(\mathbf{g}) = 1, T \mid R) &= P(T \mid R) \times P(\mathrm{CN}(\mathbf{g}) = 1 \mid T) \\ &= \prod_{\mathbf{r} \in R} \sum_{\mathbf{v} \in L^{(\mathbf{g})}} T_{\mathbf{rv}} \cdot \mathcal{P}_{\mathbf{rv}} \times \prod_{w \in W^{(\mathbf{g})}} \varphi_w(T) \end{aligned} \qquad (7)$$

Next, we moved the calculations to log-space, added window weight $\zeta_w$ and introduced the contribution factors $\Omega_R, \Omega_D \geq 0$, which represent the relative importance of read alignment and read depth likelihoods, respectively. Then, the log-likelihood $\mathcal{L}$ can be written as:

$$\mathcal{L}_T^{(\mathbf{g})} = \Omega_R \sum_{\mathbf{r} \in R} \sum_{\mathbf{w} \in L^{(\mathbf{g})}} T_{\mathbf{rw}} \log \mathcal{P}_{\mathbf{rw}} + \Omega_D \sum_{w \in W^{(\mathbf{g})}} \zeta_w \log \varphi_w(T) \qquad (8)$$

The contribution factors $\Omega_R$ and $\Omega_D$ are necessary because read alignments can overshadow read depth due to the large number of read pairs and large differences between several read alignments. The factors should sum up to two to generate the same range of likelihoods as in the unweighted case ($\Omega_R = \Omega_D = 1$). We used the default values $\Omega_R = 0.15$ and $\Omega_D = 1.85$ because they produced good results across a selection of target loci and sequencing datasets. When needed, users can provide custom $\Omega$ values to adjust read alignment and depth balance for specific loci of interest to achieve optimal accuracy.

**Likelihood update.** Given the $\mathcal{L}_T^{(\mathbf{g})}$ log-likelihood for genotype $\mathbf{g}$ and some read assignment $T$, we can efficiently calculate the $\mathcal{L}_{T'}^{(\mathbf{g})}$ log-likelihood for a new read assignment $T'$ if the read assignment has changed for only one read pair. Suppose that the read assignment changed for read pair $\mathbf{r}$ from location $uv$ (in $T$) to $u'v'$ (in $T'$). Then, the read depth likelihood values $\varphi_w(T')$ will be identical to $\varphi_w(T)$ for all windows except for $u, v, u', v'$, where read depth can be recomputed quickly. This way, the log-likelihood can be recalculated in constant time:

$$\begin{aligned} \mathcal{L}_{T'}^{(\mathbf{g})} &= \mathcal{L}_T^{(\mathbf{g})} + \Omega_R \cdot (\log \mathcal{P}_{\mathbf{r}, u'v'} - \log \mathcal{P}_{\mathbf{r}, uv}) \\ &+ \Omega_D \sum_{w \in \{u, v, u', v'\}} \zeta_w \cdot (\log \varphi_w(T') - \log \varphi_w(T)) \end{aligned} \qquad (9)$$

 

## Finding the best read assignment

For each genotype **g**, we aimed to find such read assignment $T$ that would maximize the joint $\mathcal{L}_T^{(\mathbf{g})}$ log-likelihood. Locityper implements three approaches for finding such read assignment: stochastic greedy approach[71]; simulated annealing[72]; and ILP[73]. The first two algorithms start from an arbitrarily generated read assignment $T$, then iteratively select a random read pair **r** and switch its location if it increases the genotype likelihood. In addition to good location switches, simulated annealing permits bad switches (decreasing the overall likelihood), gradually restricting the frequency of such events.

In an ILP formulation, we introduced two sets of unknowns: $x_{\mathbf{rw}} \in \{0, 1\}$ for each read pair r and each location $\mathbf{w} \in L^{(g)}$; and $y_{wd} \in \{0, 1\}$ for each window $w \in W^{(\mathbf{g})}$ and each possible window depth $d$ between zero and the maximal possible read depth ($D_{\max}$). The problem can be written as follows:

$$\text{Maximize} \quad \sum_{\mathbf{r} \in R} \sum_{\mathbf{w} \in L^{(g)}} x_{\mathbf{rw}} \cdot \Omega_R \log \mathcal{P}_{\mathbf{rw}} + \sum_{w \in W^{(\mathbf{g})}} \sum_{d=0}^{D_{\max}} y_{wd} \cdot \Omega_D \zeta_w \varphi_w(d)$$

$$\text{Subject to} \quad \sum_{\mathbf{w} \in L^{(\mathbf{g})}} x_{\mathbf{rw}} = 1 \quad \forall \mathbf{r} \in R,$$

$$\sum_{d=0}^{D_{\max}} y_{wd} = 1 \quad \forall w \in W^{(\mathbf{g})}, \tag{10}$$

$$\sum_{\mathbf{r} \in R} \sum_{u \in W^{(\mathbf{g})}} (x_{\mathbf{r},wu} + x_{\mathbf{r},uw}) - \sum_{d=0}^{D_{\max}} d \cdot y_{wd} = 0 \quad \forall w \in W^{(\mathbf{g})}$$

Note that we can remove variables $x_{\mathbf{r}}$ for trivial read pairs, which map to only one possible location; at the same time, the number of possible read depth variables $y_w$ is exactly one more than the number of nontrivial read pairs mapping to $w$. Finally, the sum $\sum_{\mathbf{r} \in R} \sum_{u \in W^{(\mathbf{g})}}$ in the third constraint can be limited to windows and read pairs relevant to window $w$. Locityper uses two commercial ILP solvers, both available under academic licenses: HiGHS[74] and Gurobi (www.gurobi.com). Note that it is possible to state a bigger ILP problem by removing the need to iterate over all possible genotypes (Supplementary Information). However, we observed that existing ILP solvers are unable to quickly and accurately find a solution to such a problem.

## Locus genotyping

To find the best locus genotype for the input WGS data, Locityper finds the best read assignment and the corresponding genotype likelihood for each possible locus genotype (Fig. 1). To speed up the process, we started by calculating the log-likelihood in the absence of read depth ($\Omega_D = 0$), which can be efficiently computed by assigning every read to its most probable location. Then, we used heuristic filtering by removing all genotypes whose likelihood is $10^{100}$ smaller than the best likelihood (the first 500 genotypes are kept regardless of the likelihood). For all remaining genotypes, the best read assignment is found using one of the three approaches described above. Even though the ILP solvers typically find better read assignments, we used simulated annealing as the default solver because it produces decent read assignments in a fraction of the ILP solving time.

Splitting locus haplotypes into nonoverlapping windows is an intrinsically discrete process. Furthermore, windows can be shifted across different haplotypes because of the presence of indels. Consequently, identical read depth profiles may produce varying read depth likelihoods depending on the window boundaries. To reduce this effect, we performed a procedure similar to noise injection regularization[75], where we randomly moved read alignment centers to either direction and reassigned reads to windows. In addition, we redefined the window GC content values and $\zeta_w$ weights as if the window was randomly moved (the actual window boundaries stay fixed). In a default configuration, read and window movement is limited to half-window size or 200 bp, whichever is smaller. Repeating noise injection several times (20 by default), together with the

stochastic nature of likelihood maximization, produces a distribution of log-likelihoods for each genotype.

Finally, Locityper selects a primary genotype with the highest average log-likelihood and calculates its Phred quality[27] based on the probability of error: the probability that the true log-likelihood of any other genotype is higher than the true log-likelihood of the primary genotype, calculated using a one-sided Welch's $t$-test[76]. Additionally, we redefined genotype probabilities as the probability of having the highest true likelihood, calculated as the product of inverse $t$-test $P$ values for all pairwise genotype comparisons.

Moreover, Locityper outputs the number of unexplained reads, which map to some but not to the two predicted haplotypes. Finally, Locityper outputs a weighted Jaccard distance between the minimizers of the primary genotype and other probable genotypes. In an unambiguous prediction, this value should be low because all likely genotypes should be similar to each other. Users can use these values for conservative post-genotyping filtering, for example, in the HiFi-based LOO evaluation; discarding 20.2% genotypes with over 50 unexplained reads raises the median QV from 36.9 to 38.2.

## Locus selection

To create a set of target loci, we started with 273 CMR genes[7]. We expanded gene coordinates to a minimum of 10 kb, when needed, and supplied positions as input to Locityper locus preprocessing, allowing an additional coordinate expansion by at most 300 kb to each of the sides (add -e 300k). At this stage, eight genes (*ATPAF2*, *CLIP2*, *GTF2I*, *GTF2IRD2*, *IGHV3-21*, *MRC1*, *NCF1* and *SMN1*) were discarded because at least one the gene ends was contained in a 300-kb-long pangenomic bubble. Afterwards, we removed redundant loci (completely contained in another locus), which produced a final set of 256 loci, containing 265 CMR genes. In similar fashion, we added 33 loci covering genes from the *MHC* and *KIR* gene clusters, and 31 loci covering the *MUC*, *CFH* and *CYP2* genes. Even though the reference panels were constructed based on 90 haplotypes from whole-genome-phased assemblies[8], on average around 80 unique haplotypes were reconstructed per locus, as some haplotypes are not unique while others are only partially assembled (Supplementary Table 1). The number of discarded haplotypes significantly correlated with genotyping accuracy: the median Locityper QV for the PacBio HiFi datasets had Spearman's $\rho = 0.67$ with the number of duplicate haplotypes ($P < 2.2 \times 10^{-16}$) and $\rho = -0.24$ with the number of unassembled haplotypes ($P = 7.5 \times 10^{-5}$).

## Data used in the study

Pangenome reference in VCF was downloaded from https://s3-us-west-2.amazonaws.com/human-pangenomics/pangenomes/freeze/freeze1/minigraph-cactus/hprc-v1.1-mc-grch38/hprc-v1.1-mc-grch38.raw.vcf.gz. Illumina, PacBio HiFi and Oxford Nanopore data for the HPRC samples can be found at https://s3-us-west-2.amazonaws.com/human-pangenomics/index.html?prefix=working. NYGC variant calls for the 1KGP samples were downloaded from http://ftp.1000genomes.ebi.ac.uk/vol1/ftp/data_collections/1000G_2504_high_coverage/working/20220422_3202_phased_SNV_INDEL_SV. The 3,202 1KGP Illumina datasets are available on the European Nucleotide Archive under accession nos. PRJEB31736 and PRJEB36890.

Simulated Illumina data were constructed using ART Illumina[30] v.2.5.8 with the parameters -ss HS25 -m 500 -s 20 -l 150 -f 15 for all phased haplotype assemblies from the HPRC project, which can be found on *Zenodo* https://doi.org/10.5281/zenodo.5826274.[77]

## Benchmarking Locityper

To evaluate haplotyping accuracy, we computed full-length alignments between actual and predicted haplotypes using the Locityper align module. Internally, it finds the longest common subsequence of $k$-mers using LCSk++[78] and completes the alignment between $k$-mer matches using the wavefront alignment algorithm[79,80]. Three $k$-mer

sizes are tried (25, 51 and 101); an alignment with the highest alignment score is returned.

Afterwards, we calculated the haplotyping error, that is, the sequence divergence between two haplotypes, calculated as the ratio between edit distance $\Delta$ and alignment size $S$ (edit distance plus the number of matches). As actual and predicted genotypes consist of two haplotypes, there are two possible actual–predicted haplotype pairings. Of the two options, we selected a pairing that produces a smaller ratio between sum edit distance and sum alignment size.

Then, we used Phred-like transformation of haplotyping error $QV = -10 \times \log_{10}(\Delta/S)$ to obtain the haplotyping QVs[28,81]. However, when two haplotypes are completely identical ($\Delta = 0$), QV becomes infinite, which poses problems for average QV calculation. For that reason, we corrected the QV definition:

$$QV = -10 \times \log_{10}\left(\frac{\max\{\Delta, 1/2\}}{S}\right) \tag{11}$$

This way, the QV difference between edit distances 0 and 1 is the same as between 1 and 2, and equals to $10 \times \log_{10}2 \approx 3$. Constants smaller than 1/2 were generally even more beneficial for Locityper benchmarking.

We considered a trio of locus genotypes concordant if one of the child haplotypes closely matches one of the maternal haplotypes and another closely matches one of the paternal haplotypes. Like the haplotyping error calculation, we iterated over eight possible combinations; selected one with the smallest sum edit distance divided by the sum alignment size; and calculated the QV score for each of the child haplotypes.

To compare Locityper with state-of-the-art PacBio HiFi pipelines, we obtained existing[8] unphased DeepVariant[32] v.1.1.0 single-nucleotide polymorphism and indel calls for the PacBio HiFi HPRC datasets, which we phased using WhatsHap[82] phase v.2.3. Next, we used the WhatsHap haplotag to assign reads to haplotypes and used Sniffles[31] v.2.4 to generate phased SVs. Finally, we used RTG[83] vcfmerge v.3.12.1 to generate the merged Sniffles + DeepVariant call set.

We used the Bcftools[84] v.1.21 consensus to reconstruct haplotypes from each of the three phased variant call sets (Sniffles, Sniffles + DeepVariant and 1KGP[3]). In the process, we removed contradicting overlapping variant calls, and variants with symbolic alternative alleles (with exception of <DEL>) because they cannot be used for haplotype reconstruction.

Finally, we used T1K[23] v.1.0.5 with the presets hla-wgs --alleleDigitUnits 15 --alleleDelimiter : and kir-wgs with all other parameters set to default. Ground-truth *HLA* and *KIR* annotation for the HPRC assemblies were obtained with Immuannot[85] using the allele databases[35,86] IPD-IMGT/HLA v.3.55 and IPD-KIR v.2.13. If a haplotype contains a new gene allele, Immuannot may associate it with several existing alleles. In such cases, we evaluated the predicted allele according to the best-matching existing allele.

In all evaluations, we used Locityper v.0.18.0 along with its dependencies SAMtools[84] v.1.21, Jellyfish[62] v.2.2.10, Strobealign[63] v.0.13.0 and Minimap2 (ref. [64]) v.2.26-r1175.

### Reporting summary

Further information on research design is available in the Nature Portfolio Reporting Summary linked to this article.

## Data availability

Locityper-predicted genotypes for 3,202 Illumina 1KGP samples, corresponding preprocessed WGS parameters, target locus database, simulation seeds and benchmarking results can be found on Zenodo[87] (https://doi.org/10.5281/zenodo.10977559). The pangenome reference in VCF was downloaded from https://github.com/human-pangenomics/hpp_pangenome_resources (GRCh38 Graph,

raw VCF). Illumina, PacBio HiFi and Oxford Nanopore data for the HPRC samples can be found at https://s3-us-west-2.amazonaws.com/human-pangenomics/index.html?prefix=working. NYGC variant calls for the 1KGP samples were downloaded from http://ftp.1000genomes.ebi.ac.uk/vol1/ftp/data_collections/1000G_2504_high_coverage/working/20220422_3202_phased_SNV_INDEL_SV. The 3,202 1KGP Illumina datasets are available on the European Nucleotide Archive under accession nos. PRJEB31736 and PRJEB36890.

## Code availability

Locityper is implemented in the Rust programming language, and can be installed via conda, singularity and docker. The source code is freely available under the terms of the MIT license at https://github.com/tprodanov/locityper along with installation and usage instructions. The Locityper v.0.18.0 source code and additional benchmarking scripts can be downloaded from Zenodo[88] (https://doi.org/10.5281/zenodo.10979046).

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

## Acknowledgements

We thank the MHC working group of the Human Genome Structural Variation Consortium for valuable feedback on an earlier version of the evaluation. This research was supported in part by funding from the National Institutes of Health National Human Genome Research Institute (grant no. R01 HG002385 to E.E.E. and T.M.) and grant no. U01 HG013748 to T.M. E.E.E. is an investigator of the Howard Hughes Medical Institute.

## Author contributions

T.P. and T.M. conceived the project and designed the algorithm. T.P. developed the software and performed the analyses. T.P. and E.G.P. prepared the figures. T.P., E.G.P., G.S., S.G.M., E.E.E. and T.M. wrote the manuscript.

## Funding

## Competing interests

E.E.E. is a scientific advisory board member of Variant Bio. The other authors declare no competing interests.

## Additional information

**Extended data** is available for this paper at https://doi.org/10.1038/s41588-025-02362-4.

**Correspondence and requests for materials** should be addressed to Timofey Prodanov or Tobias Marschall.

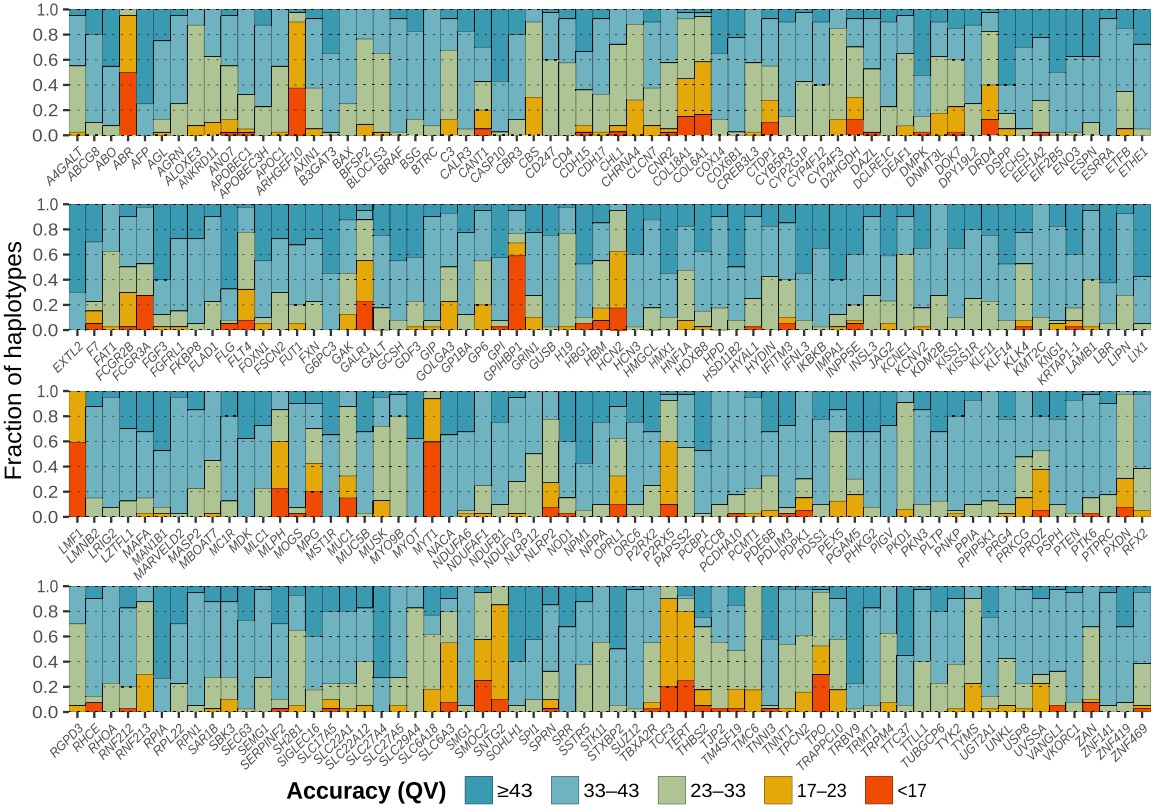

**Extended Data Fig. 1 | Locityper haplotyping accuracy across 20 PacBio HiFi datasets.** Evaluation was performed across 256 challenging medically relevant loci in leave-one-out configuration.

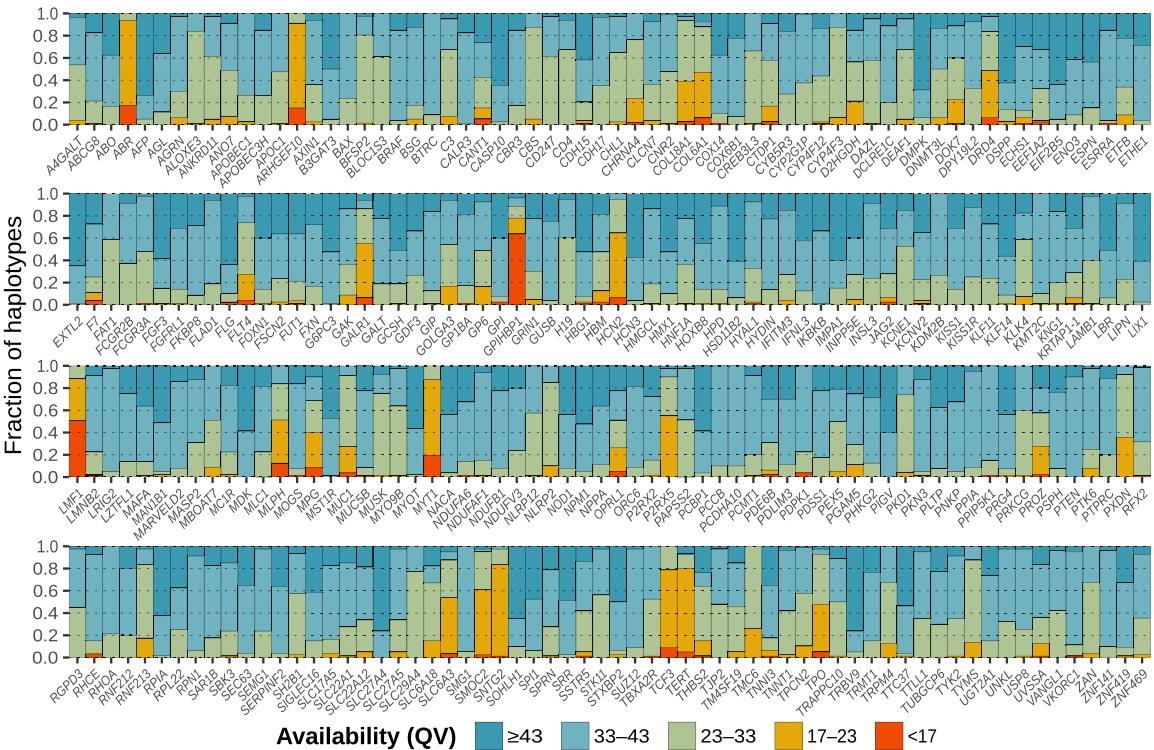

**Extended Data Fig. 2 | Haplotype availability in the leave-one-out setting.** In the leave-one-out setting, two actual sample haplotypes are removed from the database. This figure shows Phred-scaled divergence (QV) between the actual haplotypes and the closest remaining haplotypes.

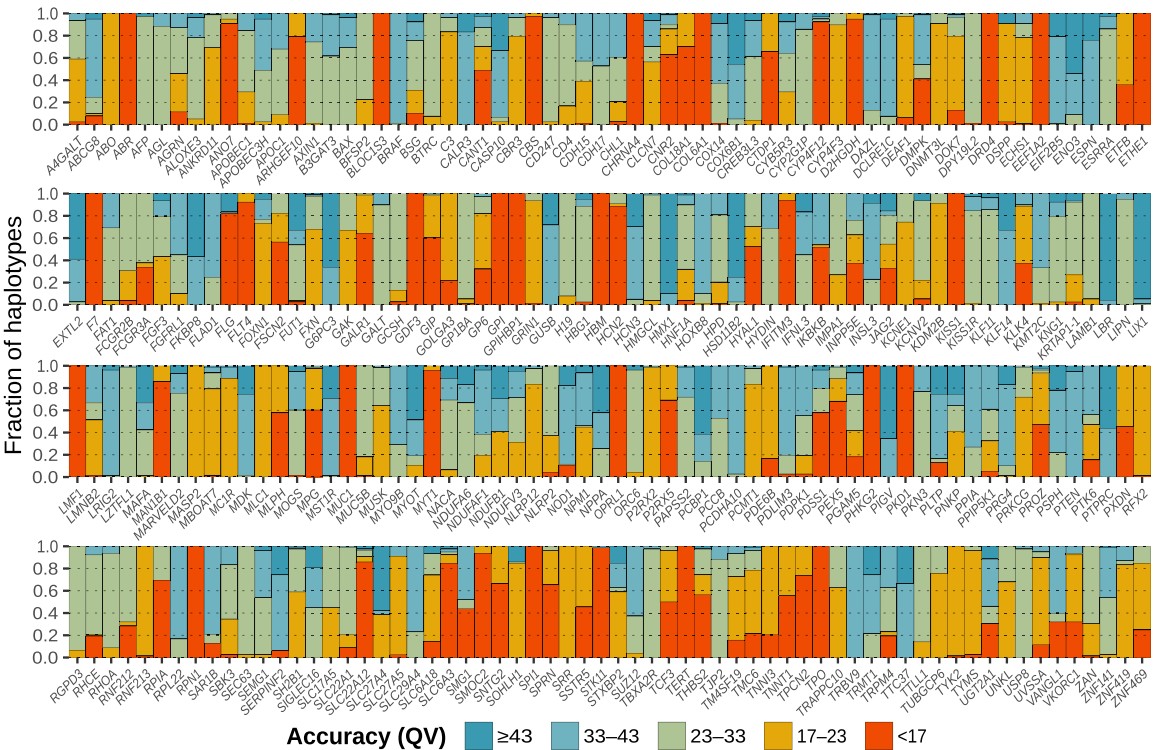

**Extended Data Fig. 3 | Accuracy of haplotypes, reconstructed from the phased 1KGP call set for 39 HPRC samples.** Accuracy is measured in QV and measured across 256 challenging medically relevant loci.

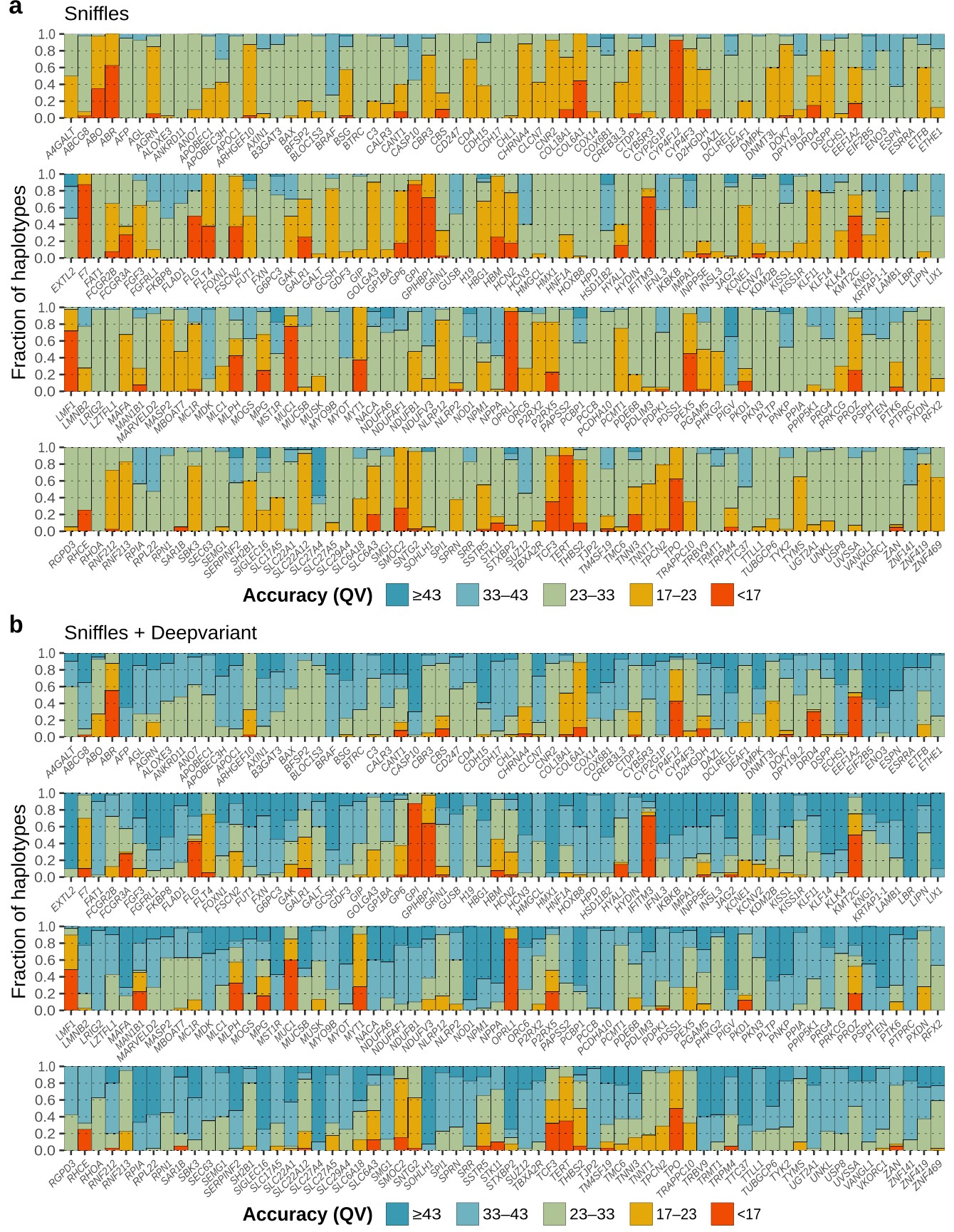

**Extended Data Fig. 4 | Sniffles haplotyping accuracy for 20 PacBio HiFi datasets.** Accuracy is calculated only for phased Sniffles calls (**a**) as well as for the merged callset of Sniffles and DeepVariant calls (**b**).

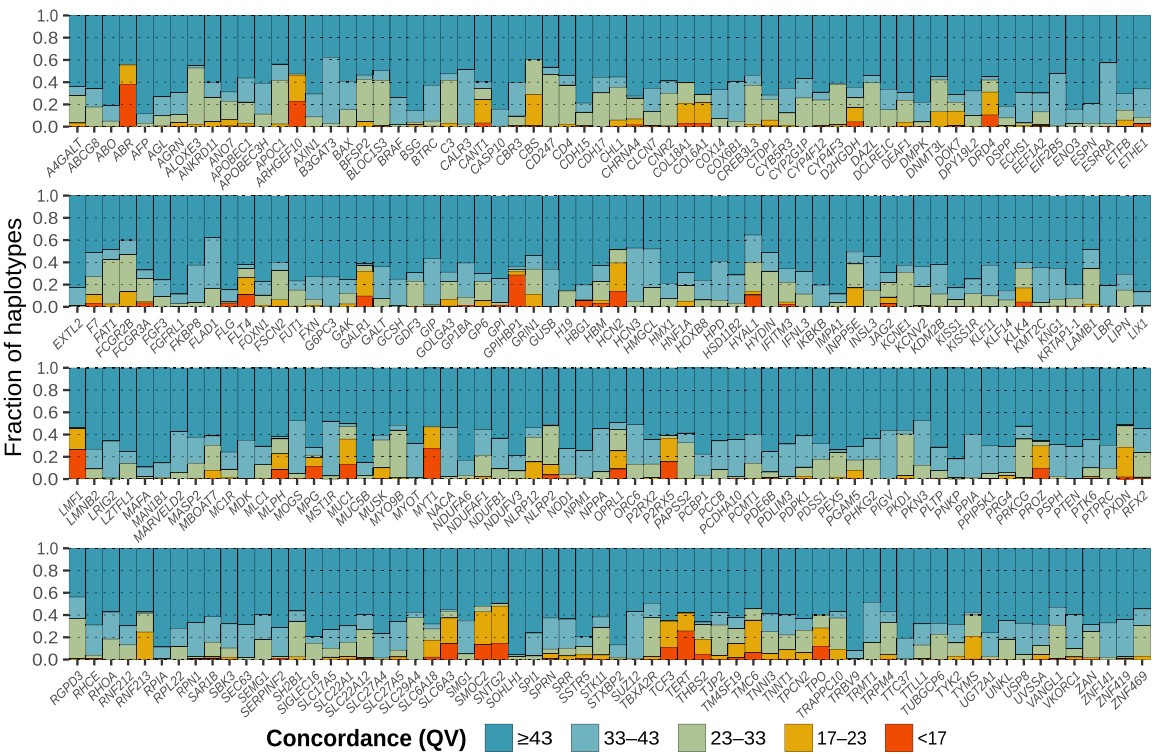

**Extended Data Fig. 5 | Locityper trio concordance.** Locityper trio concordance evaluated on Illumina WGS data for 563 trios from the 1KGP project; trios with HPRC samples were excluded.

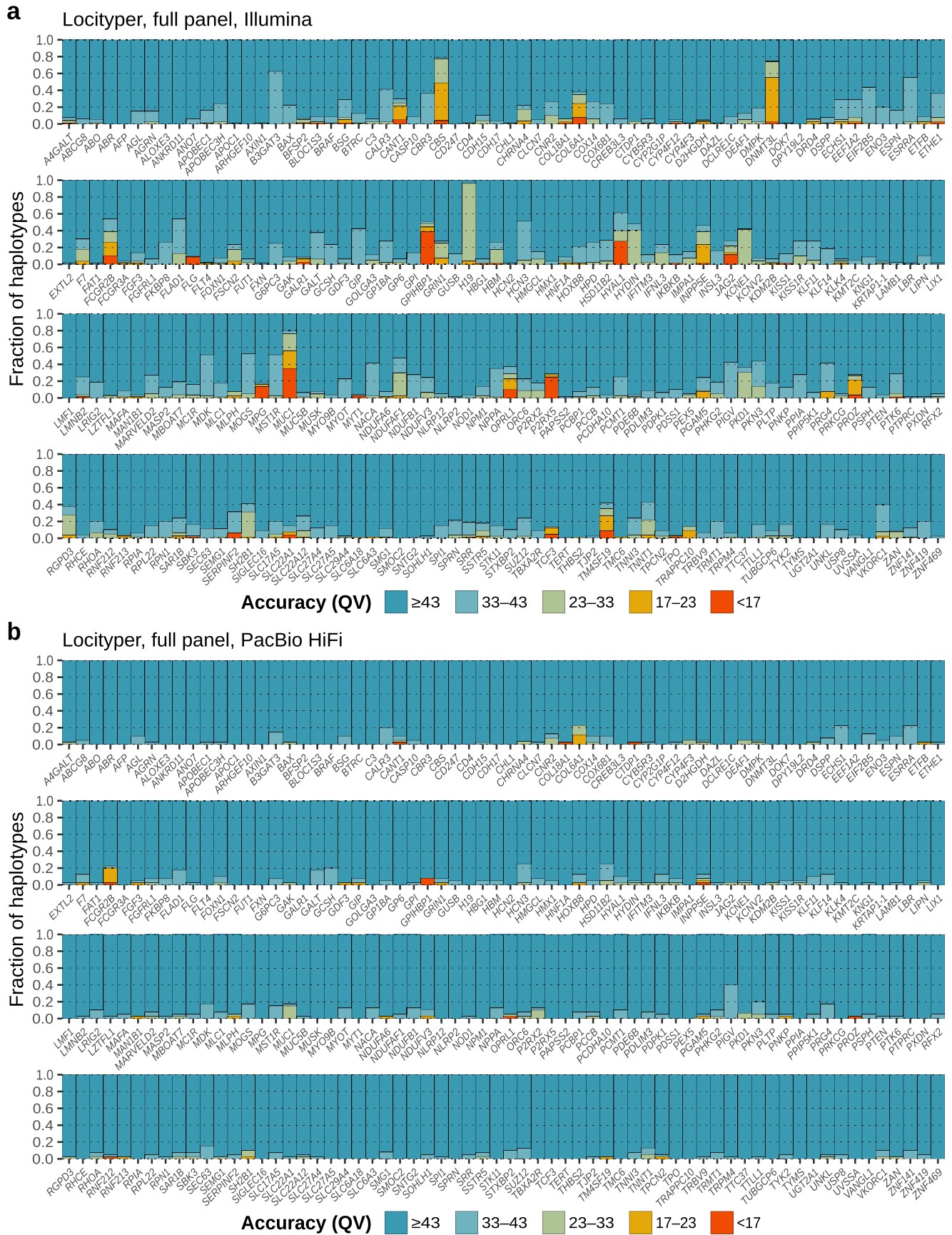

**Extended Data Fig. 6 | Locityper haplotyping accuracy using full reference panel.** Locityper haplotyping accuracy using full reference panel evaluated on 40 Illumina datasets (**a**) and 20 PacBio HiF datasets (**b**).

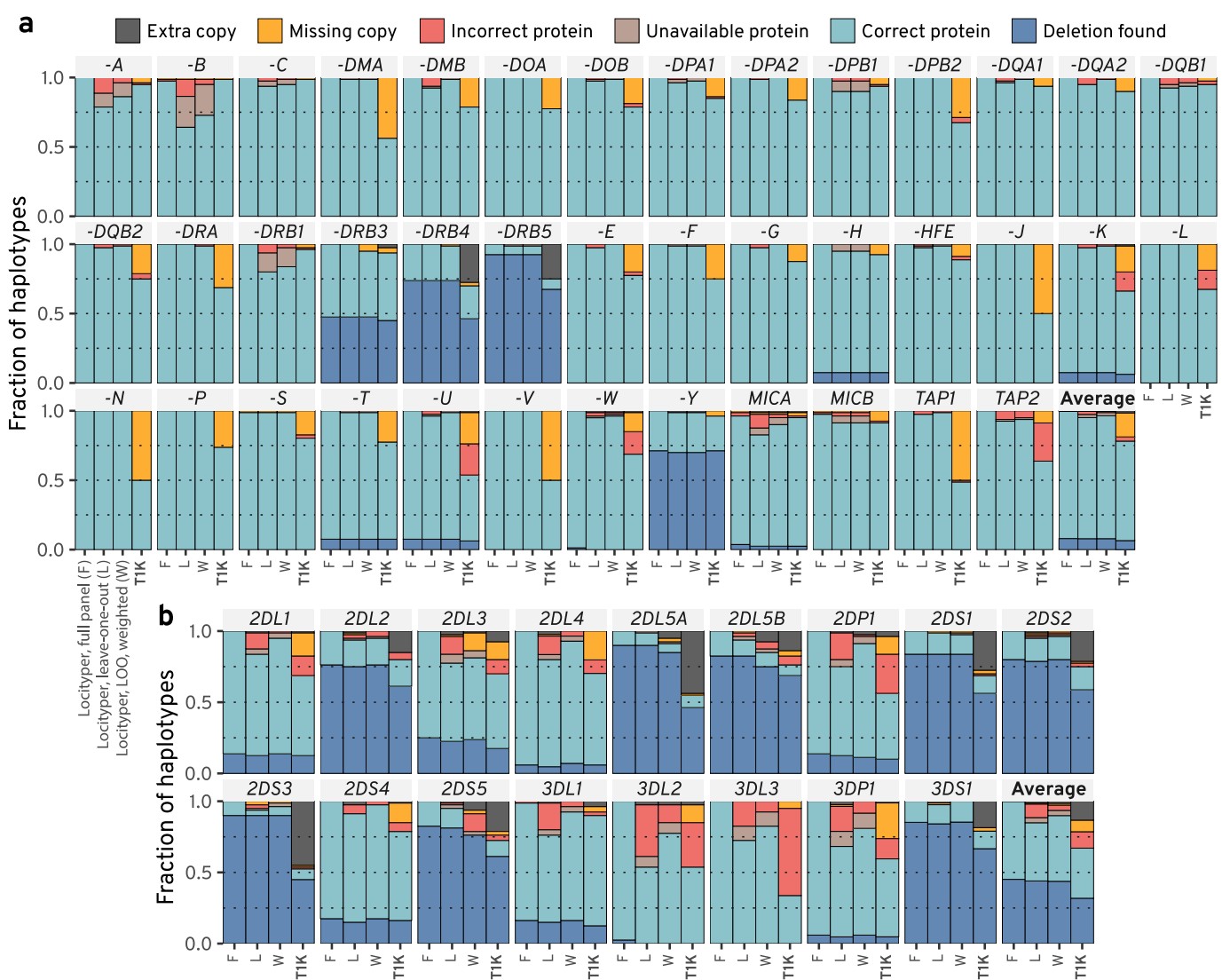

**Extended Data Fig. 7 | Stratifying predicted MHC/KIR alleles.** On each subplot, four bars represent Locityper with the full reference panel (F); Locityper in the leave-one-out setting without and with weights (denoted L and W, respectively); and T1K. Genotyping is performed for 40 HPRC samples across 40 (pseudo)genes from the MHC locus (**a**) and 17 (pseudo)genes from the KIR locus (**b**). T1K/Locityper allele predictions are placed into six categories: *extra copy* for cases when a genotyper called more gene copies than actually present in the locus; *missing copy* when a genotyper failed to call a present gene copy; *(in)correct protein* for predictions where a protein product (second field in the HLA/KIR nomenclature) was called (in)correctly; *unavailable protein* for such Locityper LOO predictions, where true protein product is unavailable in the LOO database and therefore cannot be correctly identified; and *deletion found* for cases when a genotyper correctly identified a missing gene copy. Last entry in each panel shows average fraction across all MHC/KIR genes/pseudogenes.

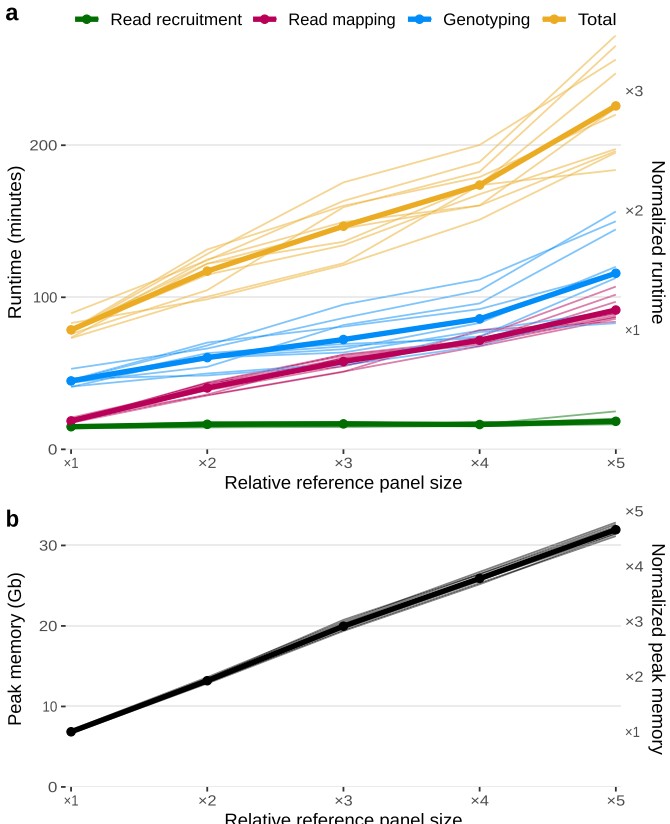

**Extended Data Fig. 8 | Locityper runtime and memory usage.** Locityper runtime and memory usage at 10 randomly selected Illumina WGS datasets and 256 target loci, with bold lines showing average values. Standard reference panel (up to 90 haplotypes) was extended with randomly mutated haplotypes to measure the effect of growing pangenomes on Locityper runtime. Correspondingly, x-axis shows reference panel size, relative to the non-extended panel. Right y-axis shows runtime/peak memory, normalized by the average (total) value at the non-extended reference panel. **a**, Runtime was measured across three non-overlapping steps: read recruitment (green); read mapping to haplotypes (red); locus genotyping (blue). Total runtime is shown in yellow. **b**, Peak memory usage (Gb).

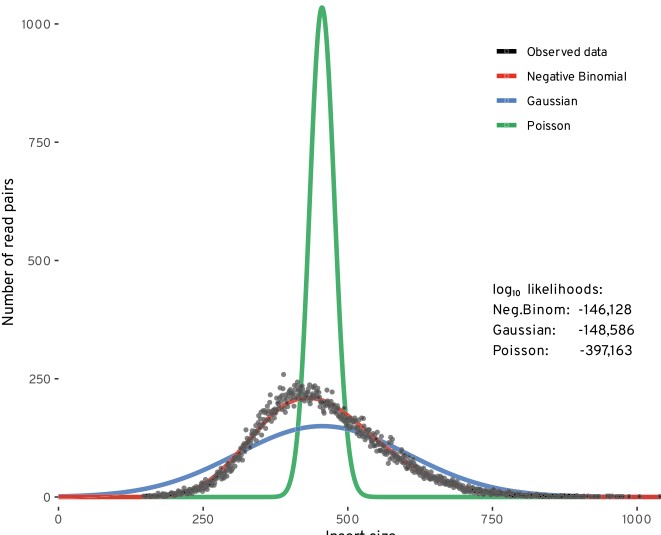

**Extended Data Fig. 9 | Insert size distribution.** Black dots show observed insert sizes for 55 thousands read pairs from the HG00621 Illumina WGS dataset. Colored lines show three fitted distributions: Negative Binomial (red), Gaussian (blue) and Poisson (green). Fit $\log_{10}$ likelihoods for all distributions are shown on the right of the figure.

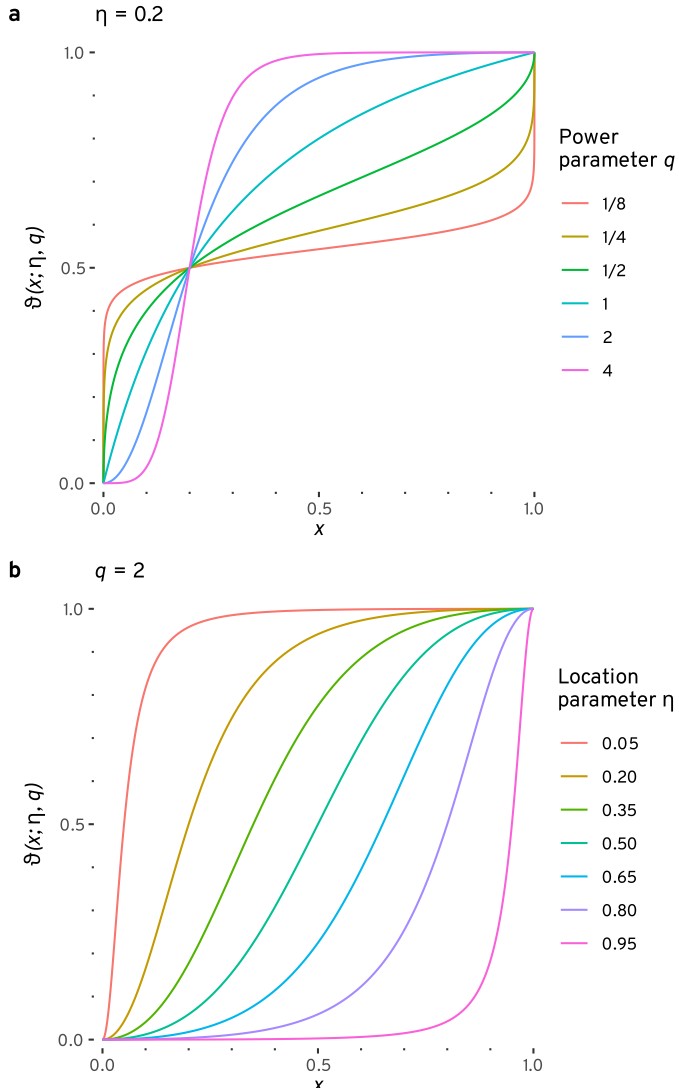

**Extended Data Fig. 10 | Two-parametric weight function $\vartheta$.** $\vartheta(x; \eta, q)$ with variable q and fixed $\eta = 0.2$ (**a**); and with variable $\eta$ and fixed $q = 2$ (**b**).

# Reporting Summary

## Statistics

For all statistical analyses, confirm that the following items are present in the figure legend, table legend, main text, or Methods section.

| n/a | Confirmed | |
|---|---|---|
| ☐ | ☒ | The exact sample size (*n*) for each experimental group/condition, given as a discrete number and unit of measurement |
| ☒ | ☐ | A statement on whether measurements were taken from distinct samples or whether the same sample was measured repeatedly |
| ☐ | ☒ | The statistical test(s) used AND whether they are one- or two-sided *Only common tests should be described solely by name; describe more complex techniques in the Methods section.* |
| ☒ | ☐ | A description of all covariates tested |
| ☒ | ☐ | A description of any assumptions or corrections, such as tests of normality and adjustment for multiple comparisons |
| ☐ | ☒ | A full description of the statistical parameters including central tendency (e.g. means) or other basic estimates (e.g. regression coefficient) AND variation (e.g. standard deviation) or associated estimates of uncertainty (e.g. confidence intervals) |
| ☐ | ☒ | For null hypothesis testing, the test statistic (e.g. *F*, *t*, *r*) with confidence intervals, effect sizes, degrees of freedom and *P* value noted *Give P values as exact values whenever suitable.* |
| ☒ | ☐ | For Bayesian analysis, information on the choice of priors and Markov chain Monte Carlo settings |
| ☒ | ☐ | For hierarchical and complex designs, identification of the appropriate level for tests and full reporting of outcomes |
| ☒ | ☐ | Estimates of effect sizes (e.g. Cohen's *d*, Pearson's *r*), indicating how they were calculated |

*Our web collection on statistics for biologists contains articles on many of the points above.*

## Software and code

Policy information about availability of computer code

| | |
|---|---|
| Data collection | ART Illumina v2.5.8 |
| Data analysis | Locityper v0.18.0 (https://github.com/tprodanov/locityper, https://zenodo.org/records/14861388), Pangenie v3.02, T1K v1.0.5, Jellyfish v2.2.10, Minimap2 v2.26-r1175, Strobealign v0.13.0, Samtools v1.21, Bcftools v1.21, Tabix v1.21, Vt v0.57721, RTG-tools v3.12.1, Immuannot e8da19c |

For manuscripts utilizing custom algorithms or software that are central to the research but not yet described in published literature, software must be made available to editors and reviewers. We strongly encourage code deposition in a community repository (e.g. GitHub). See the Nature Portfolio guidelines for submitting code & software for further information.

## Data

Policy information about availability of data

All manuscripts must include a data availability statement. This statement should provide the following information, where applicable:
- Accession codes, unique identifiers, or web links for publicly available datasets
- A description of any restrictions on data availability
- For clinical datasets or third party data, please ensure that the statement adheres to our policy

Locityper-predicted genotypes for 3202 Illumina 1KGP samples, corresponding preprocessed WGS parameters, target loci database, simulation seeds and benchmarking results can be found on Zenodo (zenodo.org/records/14861498). Pangenome reference in a variant calling format (VCF) was downloaded from

## Research involving human participants, their data, or biological material

Policy information about studies with human participants or human data. See also policy information about sex, gender (identity/presentation), and sexual orientation and race, ethnicity and racism.

| | |
|---|---|
| Reporting on sex and gender | N/A |
| Reporting on race, ethnicity, or other socially relevant groupings | N/A |
| Population characteristics | N/A |
| Recruitment | N/A |
| Ethics oversight | N/A |

Note that full information on the approval of the study protocol must also be provided in the manuscript.

# Field-specific reporting

Please select the one below that is the best fit for your research. If you are not sure, read the appropriate sections before making your selection.

☒ Life sciences       ☐ Behavioural & social sciences       ☐ Ecological, evolutionary & environmental sciences

For a reference copy of the document with all sections, see nature.com/documents/nr-reporting-summary-flat.pdf

# Life sciences study design

All studies must disclose on these points even when the disclosure is negative.

| | |
|---|---|
| Sample size | Reference panel of 90 haplotypes were used. The size equals the size of the latest (at the moment of submission) HPRC pangenome with 90 phased diploid whole genome assemblies, from where local haplotypes were taken (44 diploid samples + 2 reference assemblies). Direct evaluation was performed for all 40 HPRC samples (80 haplotypes) with available Illumina data; same 40 samples were used for simulated Illumina data. For data storage reasons, long read analysis (PacBio HiFi and ONT) was performed on 20 samples (40 haplotypes). 1KGP call set comparison was performed on all 39 samples with both HPRC assemblies and NYGC diploid calls. Trio concordance was calculated on all 563 trios (1676 samples) from the 1KGP cohort, independent from the HPRC cohort. All available samples were used, except for long read analysis. Sample size of 40 is generally considered sufficient for basic statistical analysis; additionally, any random effects should be almost fully offset by leave-one-out analysis and by large sample-size trio analysis. |
| Data exclusions | No data exclusion. |
| Replication | Every samples was analysed twice, with full reference panel and with limited leave-one-out panel to model real life independence between reference panels and analyzed samples. No replications within each analysis was needed since all tools are either deterministic (same analysis produces the same results), or have random elements but produce virtually the same results evrey time. All performed analyses and replications were included in the manuscript or in supplementary information. |
| Randomization | For long read data, 20 samples were selected randomly. Elsewhere: all available data was used, no allocation needed. |
| Blinding | Samples were not grouped into case-control, instead the only relevant information could be the similarity between analyzed sample haplotypes and other haplotypes. This information was not used by researchers until the evaluation stage. Furthermore, the analysis was performed automatically using Lociyper, which does not support input similarity matrix, and therefore could not be influenced by it. |

# Reporting for specific materials, systems and methods

We require information from authors about some types of materials, experimental systems and methods used in many studies. Here, indicate whether each material, system or method listed is relevant to your study. If you are not sure if a list item applies to your research, read the appropriate section before selecting a response.

## Materials & experimental systems

| n/a | Involved in the study |
|-----|----------------------|
| ☒ | Antibodies |
| ☒ | Eukaryotic cell lines |
| ☒ | Palaeontology and archaeology |
| ☒ | Animals and other organisms |
| ☒ | Clinical data |
| ☒ | Dual use research of concern |
| ☒ | Plants |

## Methods

| n/a | Involved in the study |
|-----|----------------------|
| ☒ | ChIP-seq |
| ☒ | Flow cytometry |
| ☒ | MRI-based neuroimaging |

## Plants

| | |
|---|---|
| Seed stocks | *Report on the source of all seed stocks or other plant material used. If applicable, state the seed stock centre and catalogue number. If plant specimens were collected from the field, describe the collection location, date and sampling procedures.* |
| Novel plant genotypes | *Describe the methods by which all novel plant genotypes were produced. This includes those generated by transgenic approaches, gene editing, chemical/radiation-based mutagenesis and hybridization. For transgenic lines, describe the transformation method, the number of independent lines analyzed and the generation upon which experiments were performed. For gene-edited lines, describe the editor used, the endogenous sequence targeted for editing, the targeting guide RNA sequence (if applicable) and how the editor was applied.* |
| Authentication | *Describe any authentication procedures for each seed stock used or novel genotype generated. Describe any experiments used to assess the effect of a mutation and, where applicable, how potential secondary effects (e.g. second site T-DNA insertions, mosiacism, off-target gene editing) were examined.* |

