## [Peer Review file · Nature Genetics]

Locityper enables targeted genotyping of complex polymorphic genes

Corresponding Author: Professor Tobias Marschall

Version 0:

Decision Letter:

19th Sep 2024

Dear Tobias,

Firstly, my extensive and profound apologies for the delay in the review - as previously conveyed, we needed to secure a replacement referee for the two who had stopped responding to emails at a very late stage of review. Thankfully they submitted extremely speedily and we have sufficient feedback at this point for a decision.

Your Technical Report, "Locityper: targeted genotyping of complex polymorphic genes" has now been seen by 2 referees. You will see from their comments below that while they find your work of interest, some important points are raised. We are interested in the possibility of publishing your study in Nature Genetics, but would like to consider your response to these concerns in the form of a revised manuscript before we make a final decision on publication.

Briefly, the two reviews both sound appreciative on the potential advance of utility of Locityper.

Referee #1 says that your tool could be useful, but suggests the benchmarking and analyses do not support the claims made in your work; they also note some potential technical issues that may be affecting performance (in e.g. HLA typing). They do, however, provide thoughtful suggestions to address the issues raised.

Reviewer #2 is more straightforwardly enthusiastic, but also has a number of major comments, some of which overlap with the more critical review (e.g. on putting the more realistic LOO benchmarking upfront).

In our reading of these reports the referees have provided useful guidance and none of the requested work seems unduly difficult; we thus hope you and your co-authors will be able to address them fully. We'd particularly highlight that shared comment (on the LOO scenario), as well as Referee #1's points on HLA/KIR genotyping - these are important use-cases for Locityper, especially HLA, and we think it would substantially broaden the audience for your work if you can show clear improved performance.

To guide the scope of the revisions, the editors discuss the referee reports in detail within the team, including with the chief editor, with a view to identifying key priorities that should be addressed in revision and sometimes overruling referee requests that are deemed beyond the scope of the current study. We hope that you will find the prioritized set of referee points to be useful when revising your study. Please do not hesitate to get in touch if you would like to discuss these issues further.

We therefore invite you to revise your manuscript taking into account all reviewer and editor comments. Please highlight all changes in the manuscript text file. At this stage we will need you to upload a copy of the manuscript in MS Word .docx or similar editable format.

*2) If you have not done so already please begin to revise your manuscript so that it conforms to our Technical Report format instructions, available

[here](http://www.nature.com/ng/authors/article_types/index.html).

*3) Include a revised version of any required Reporting Summary: <https://www.nature.com/documents/nr-reporting-summary.pdf>

Please be aware of our [guidelines](https://www.nature.com/nature-research/editorial-policies/image-integrity) on digital image standards.

Link Redacted

Sincerely,

Michael Fletcher, PhD
Senior Editor, Nature Genetics
ORCID: 0000-0003-1589-7087

Referee expertise: genome bioinformatics, genetics.

Reviewers' Comments:

Reviewer #1 (Remarks to the Author):

In this manuscript, the authors describe a computational method Locityper to infer the haplotypes of any genes from the HPRC resource using a WGS sample. Locityper supports both short-read and long-read WGS data. The authors designed a likelihood function and an ILP model to find the most likely read assignments and the genotype. The authors then conducted several applications to show that Locityper was accurate even when the true haplotype may be missing in the reference haplotype. Locityper is a useful tool that can leverage HPRC resources to investigate many genes, including hundreds of challenging medically relevant genes. However, the claims and evaluations in the manuscript are biased and could not reflect the accuracy of Locityper in general applications. The following are the issues.

1. The evaluation using Locityper with the full reference database on WGS data sets overlapping with HPRC samples is highly biased. With the full reference database, the truth is guaranteed in the reference and the accuracy (QV score) is expected to be high. The author emphasizes too much on the ideal case, including emphasizing it in the abstract. Because a similar haplotype sequence of a gene may be present in another HPRC sample, Locityper-LOO is more realistic and could be explored more. For example, the QV score from Locityper-LOO might be much lower than the Locityper, but the haplotype it found might be the sequence with the highest QV score in the LOO reference haplotypes. This would suggest that Locityper-LOO still selects the optimal haplotype. For the same reason, the Locityper-LOO may be included in the comparison against Pangenie and 1KGP for the variant calls evaluation.

2. Locityper-LOO's performance on HLA-A, HLA-B, and HLA-C genes is not good. It is expected that the accuracy of Locityper-LOO at 2-field, 3-field, and full-match will be lower, as there might be one representative allele in the HPRC. However, the 1-field match should still be good. For example, for HLA-A, only HLA-A*25 and HLA-A*32 show up once in the HPRC samples based on Immuanot's prediction. For HLA-B, only HLA-B*54 and HLA-B*56 show up once, and the

majority of the HLA-B's first field shows up in more than one sample. However, Locityper-LOO's still obtained much worse accuracy at the 1-field match, e.g., about 75% for HLA-B. One potential explanation is that Locityper considers intronic region sequences, while HLA field definition is based on protein-coding sequences. If this is the case, it may suggest that Locityper's current implementation needs to be adjusted for HLA genotyping for clinical applications.

3. It seems Locityper only genotypes KIR group A haplotype genes, and the 8 group B haplotype genes, like KIR2DL2, are not processed. Therefore, Locityper still needs extra input to support KIR genotyping.

4. The authors mention that Locityper is a highly efficient method. However, the method relies on integer linear programming, which is slow, and infers the likelihood for every genotype scenario. This may suggest that Locityper's time complexity for one locus is at least $O(|H|^2)$, where $|H|$ is the number of haplotypes for a locus. In the manuscript and current HPRC, $|H|$ seems to be 90, which is a small number. HPRC aims to release 350 people's assemblies in the future, which will increase $|H|$ to 700. The authors may conduct experiments like varying $|H|$ from 20 to 90 to see how long the time is needed on the genotyping step (excluding the time on steps like read recruitment). This may give an idea of Locityper's time efficiency when HPRC evolves.

5. The comparison of Locityper with T1K could be unfair, as they are using very distinct reference sequences. It might be unfair for Locityper on HLA-A, -B, and -C genes that are well curated in the IMGT/IPD-HLA database which T1K uses. It might be unfair for T1K in other genes like HLA-J where HPRC might provide better representation than IMGT/IPD-HLA. Since IMGT/IPD-HLA and IPD-KIR contain the genome region sequence, the authors may consider using them as the reference in Locityper to show that Locityper is competitive in HLA or KIR genotyping.

On the other hand, T1K can also take the plain genome sequence in FASTA format as the reference sequence. Though T1K is not tuned to genotype all the CMR genes with the full intronic sequences, the authors may consider testing T1K's performance using the loci genome sequences extracted by Locityper as the reference FASTA file. This comparison may demonstrate that Locityper's computational model is better for this task and address the claim "Even though these methods can achieve high accuracy, they typically rely on specific gene structure and cannot be easily scaled to include more targets." In the introduction section.

6. Some KIR genes and HLA genes may be totally missing in a person, and Locityper correctly predicted that from the sequencing data. The description of the method does not explicitly model the case of missing a gene on both chromosomes. Is capturing gene missing through the filter using the potentially incorrect prediction flag?

7. The method section is a bit confusing. The joint likelihood is based on the equation $P(g,T|R)=P(g|T)P(T|R)$ below (3). However, $P(T|R)$ defined in (3) also depends on g through the term $L_r(g)$. The formulation of the likelihood function may need some clarification.

Reviewer #2 (Remarks to the Author):

This interesting work describes Locityper, a software tool designed for genotyping highly polymorphic loci in the human genome using both short- and long-read sequencing data. The authors address significant challenges in resolving genomic regions that are typically difficult to analyze due to structural complexity. Their basic idea is to optimize the alignments of *read sets* to haplotype pairs. Unlike other approaches that rely on alignments optimized for each read individually, Locityper optimizes alignments of all reads at a locus simultaneously. Results convincingly show the approach's effectiveness, significantly expanding the number of challenging medically-relevant genes available for genotyping and outperforming specialized software for HLA and KIR genes. It is striking to see that that using pangenome haplotypes (enhanced new data) can lead to major genotyping improvements while the use of genome graphs (concepts for data representation) has only led to marginal genotyping improvements so far. This is in line with previous genotyping results using the T2T genome compared to older reference genomes. Overall, I found the manuscript to be of high quality, the authors address an important challenge, results are impressive, and the software tool has already been used in other projects (DOI 10.1016/j.ajhg.2024.06.007S and 10.1101/2024.04.18.590093).

I would like to raise the following four major points:

(1) The results should focus on the "leave-one-out" setting rather than the "full-database" setting, i.e., further emphasize the "leave-one-out" experiment. Locityper was exclusively tested on HPRC datasets using HPRC reference haplotypes in a "full-database" and a "leave-one-out" setting. In the "full-database" setting, the true haplotypes of the genotyped genomes are part of the reference haplotype set. This is useful for establishing a baseline in an ideal setting. In the "leave-one-out" setting, the true haplotypes are removed from the reference set. The "leave-one-out" setting mimics real application scenarios, where the true haplotype is not guaranteed to be part of the reference set. Figure 3 currently presents the (overwhelming) results for the full database, whereas the leave-one-out results (still impressive compared to the 1KGP call set) are hidden in Suppl. Fig. 2. In addition, the comparison to the 1KGP call set in the Pangenie section is unfair as both Locityper and Pangenie were run with the full database.

(2) Along the same lines, the authors use a measure they term "lost accuracy" (e.g. Fig. 2f) for evaluating the "leave-one-out" results. While the main text defines the "lost accuracy" as the "difference between best possible and predicted QVs", the

Methods section on page 23 reveals a modification of this difference for "very good possible haplotypes". I found out about the modification only by digging deep into the Methods section, there is no hint to it in the main text. I strongly recommend finding another way to separate the error for "very good possible haplotypes" from the error for "less good possible haplotypes".

(3) While the evaluation of a 1KGP call set nicely establishes baseline performance of a standard pipeline for short-read data (Page 6), the benefit of using Locityper on long-read data compared to a standard pipeline remains unclear. Long read data simplifies analysis of many complex genomic loci. How does a simple standard pipeline (e.g. minimap2+sniffles) compare to Locityper on long reads?

(4) The authors advertise a fast version of Locityper requiring less than 6 minutes for HLA and KIR genotyping. This will be of tremendous interest for large sequencing projects with already aligned WGS data where computational time is expensive. Is this version similarly accurate? More generally, what is the impact of read recruitment accuracy on genotyping accuracy?

Minor points for further improving the manuscript:

While the basic idea of the method in Locityper is rather simple, its implementation involved choosing many distributions, parameters and thresholds. I am wondering how robust the choice of these variables are. What are the variables that influence the results the most apart from the Omegas?

The discussion section highlights strong and weak points of the approach (e.g., its departure from the variant-centric approach and excellent agreement on trio data as well as the limited size of the currently available haplotype panels or challenges in genotyping loci with significant homology to other parts of the genome), thereby providing valuable guidance on what a user can expect from Locityper now and in the future. Is it possible to apply Locityper on any canonical or complex, biallelic or multiallelic variant? Would it be possible and beneficial to use it genome-wide for SV genotyping?

Page 3 "We show that finding a maximum likelihood read assignment can be formulated as an integer linear programming (ILP) problem (Methods), for which Locityper employs existing ILP solvers and stochastic optimization."
This sentence is misleading. In my understanding, Locityper does not employ stochastic optimization for solving the ILP but it either solves an ILP or employs stochastic optimization for finding a maximum likelihood read assignment.

Page 4 "For each target locus we used a reference panel of up to 90 haplotypes"
What does "up to" mean? Please provide mean and variance of number of haplotypes that remain after preprocessing target loci. Does the number of haplotypes correlate with genotyping accuracy?

Page 5 "consequently, we will split genotypes into those that failed and those that passed filtering."
Please reference Methods section.

Page 5 "out of a total of 20,350 fully assembled sample-locus haplotypes across the 256 CMR loci and 40 Illumina WGS samples"
Why 20,350 in total? $256 \times 2 \times 40 = 20480 \neq 20350$

Page 7 "In addition to the HPRC dataset, we genotyped the full 1KGP cohort of 3202 Illumina WGS samples, including 602 trios"
Does this cohort include the 39 HPRC samples that is mentioned to overlap the 1KGP cohort on page 6? If yes, do you see better concordance for these samples, where the haplotype is part of the reference haplotypes?

Page 12 "Locityper employs integer linear programming and stochastic optimization"
Should read "Locityper employs integer linear programming or stochastic optimization".

Page 17 "A read pair is retained if both read ends have at least one good alignment ($p \geq 0.01$) to at least one of the haplotypes. All alignments with $BB_p < 0.001$ are discarded" and Page 20
How many alignments per read are retained per haplotype pair on average? How big is the search space for optimizing read assignment? Does this number somehow influence the predicted genotype's accuracy?

Page 17 "For null windows, we define insert size probability as the highest probability achievable under the precomputed insert size distribution."
Why the highest?

Page 17 "Finally, we will denote the full set of possible read pair locations on haplotype h as $L(h)_r \subset W(h) \times W(h)$ "
Why do you use L_r instead of just L ? Did I miss that L somehow depends on r ?

Page 19 "The two values need to be defined in advance and should sum up to 2"
Why do they need to sum up to 2? Don't they scale the likelihoods for all haplotype pairs in the same way so that you will obtain the same results even if they don't sum up to 2?

Page 21 "weighted by the corresponding genotype probabilities"
How do you compute genotype probabilities? Do you normalize the genotype likelihoods?

Page 21 "we marked genotypes as potentially incorrect if weighted distance is over 30"

Can you use a fixed threshold across all loci? Doesn't this depend on the similarity of the haplotypes in the reference set?

Supp. Table 1

Please indicate the reference genome for the given coordinates.

Version 1:

Decision Letter:

Our ref: NG-TR65646R

21st May 2025

Dear Dr. Marschall,

Thank you for submitting your revised manuscript "Locityper: targeted genotyping of complex polymorphic genes" (NG-TR65646R). It has now been seen by the original referees and their comments are below. The reviewers find that the paper has improved in revision, and therefore we'll be happy in principle to publish it in Nature Genetics, pending minor revisions to satisfy the referees' final requests and to comply with our editorial and formatting guidelines.

Sincerely,
Wei

Wei Li, PhD
Senior Editor
Nature Genetics
www.nature.com/ng

Reviewer #1 (Remarks to the Author):

The authors have addressed my concerns. Happy to see the review is helpful for the locityper wegithed mode.

A minor comment about "T1K often predicted smaller copy number than required" in the HLA/KIR benchmark. This is because T1K by default does not report copy number variations, so homozygous HLA/KIR alleles will only report as one. If copy number is needed, users may use "t1k-copynumber.py" in the T1K package to infer the copy number. This script is described in the "immuannot"'s publication, which the author also used in the evaluations. The authors have done a nice comparison in Figure S8 that can handle the copy number issue, so this comment is minor and just to provide some background information.

Reviewer #2 (Remarks to the Author):

The authors responded to all issues raised by me and the other reviewer and appropriately addressed all my major concerns by focussing on the more realistic LOO experiment and adding further assessments showing Locityper's superior performance. Again, I found the manuscript to be of high quality and the results are excellent. All remaining points are minor.

While the authors *replied* to all my initial minor comments, comments 2.5, 2.6, 2.8, 2.13 and 2.16 have not been addressed by making changes to the manuscript. This is not critical but unfortunate as the responses contain insights that I consider interesting for readers.

For example, in response to comment 2.8 the authors computed a significant negative correlation between accuracy and number of available haplotypes. In my understanding, the correlation will strengthen their statement that a "growing number of haplotypes in pangenomes are likely to increase Locityper accuracy even further" if mentioned in the manuscript.

Comment 2.19: I still could not find the reference genome build used.

In caption of Figure 3 "Corresponding haplotype availability can be found in Supp. Figure 2": Misled by the response to my comment 2.10, I expected to find the number of fully assembled haplotypes. Instead I found the divergence to the closest remaining haplotype. This is fine and defined further below in the manuscript but I suggest rephrasing the sentence in the caption to be more clear.

can be misleading due to potential differences in variant representation, which would likely be even more pronounced in a LOO setting. We think that comparing the 1kGP set to Locityper LOO in the haplotype-centric evaluation presented in Figure 3 is the correct way least affected by alignment artifacts and differences in representation.

Comment 1.2: Locityper-LOO's performance on HLA-A, HLA-B, and HLA-C genes is not good. It is expected that the accuracy of Locityper-LOO at 2-field, 3-field, and full-match will be lower, as there might be one representative allele in the HPRC. However, the 1-field match should still be good. For example, for HLA-A, only HLA-A*25 and HLA-A*32 show up once in the HPRC samples based on Immuannot's prediction. For HLA-B, only HLA-B*54 and HLA-B*56 show up once, and the majority of the HLA-B's first field shows up in more than one sample. However, Locityper-LOO's still obtained much worse accuracy at the 1-field match, e.g., about 75% for HLA-B. One potential explanation is that Locityper considers intronic region sequences, while HLA field definition is based on protein-coding sequences. If this is the case, it may suggest that Locityper's current implementation needs to be adjusted for HLA genotyping for clinical applications.

Response: We thank the reviewer for this important comment. We have now added a weighted mode to Locityper, where users can supply custom weights across the target haplotypes. In this instance, we downweighted introns by 0.01 and intergenic sequence by 0.005. This raised Locityper accuracy by 2% (full match) and 1.4% (protein product) on average at the MHC locus, and by 5.4% and 5% at the KIR gene cluster (see Figure 4). At the Class I MHC genes, in the weighted mode 2-field accuracy improved from 78.8% to 86.2% (*HLA-A*); 64.2% to 72.8% (*HLA-B*); and 93.8% to 95% (*HLA-C*) (see Figure 4). At the same time, complete mismatch cases significantly decreased: 8.8% → 2.5%; 23.5% → 11.1%; 2.5% → 0.0% at *HLA-A*, *-B* and *-C*, respectively. Notably, in the weighted mode Locityper achieves higher 1-field accuracy than T1K at *HLA-A* and *HLA-C* (97.5% against 96.2% and 100% against 98.8%).

Additionally, we observed that for the Class I MHC genes, 73–82% of 2-field errors are explained by unavailable alleles (see Supp. Figure 8), suggesting that the size of the reference database is presently still a limitation. Nevertheless, we emphasize that Locityper can competitively genotype even the most difficult loci, and the full database experiments suggest that the accuracy will continue to improve with larger reference panels.

Figure 4. Haplotyping accuracy for 40 HPRC samples at the MHC and KIR loci. Subpanels show fraction of haplotypes, predicted with varying accuracy at 40 (pseudo)genes from the MHC locus (a) and 17 (pseudo)genes from the KIR gene cluster (b). Fully predicted alleles, as well as correctly identified missing copies, are colored with dark green (*Full match*), due to the different number of allele fields in the HLA/KIR gene nomenclature^{37,39}. Otherwise, haplotypes are colored according to the number of correctly predicted fields. Accuracy is shown for Locityper with the full reference panel (F); Locityper in the leave-one-out setting without and with weights (denoted L and W, respectively); and T1K. Last entry in each panel shows average accuracy across all corresponding genes/pseudogenes.

Comment 1.3: It seems Locityper only genotypes KIR group A haplotype genes, and the 8 group B haplotype genes, like KIR2DL2, are not processed. Therefore, Locityper still needs extra input to support KIR genotyping.

Response: We thank the reviewer for noticing our oversight. We now fixed the issue and extended our analysis to 4 more MHC (pseudo)genes and 8 more KIR genes. At these genes, Locityper achieved even higher accuracy than on the initial gene set: average 2-field accuracy 97.8% against 96.3% at the MHC locus and 91.1% against 88.9% at the KIR genes.

Comment 1.4: The authors mention that Locityper is a highly efficient method. However, the method relies on integer linear programming, which is slow, and infers the likelihood for every genotype scenario. This may suggest that Locityper's time complexity for one locus is at least $O(|H|^2)$, where $|H|$ is the number of haplotypes for a locus. In the manuscript and current HPRC, $|H|$ seems to be 90, which is a small number. HPRC aims to release 350 people's assemblies in the future, which will increase $|H|$ to 700. The authors may conduct experiments like varying $|H|$ from 20 to 90 to see how long the time is needed on the genotyping step (excluding the time on steps like read recruitment). This may give an idea of Locityper's time efficiency when HPRC evolves.

Response: We thank the reviewer for this comment and agree that this is an important question to clarify. Enumerating all pairs of haplotypes is indeed a quadratic procedure, and because of that we perform initial filtering by examining read alignments to each genotype without evaluating read depth. This procedure is still quadratic, but very fast compared to actual likelihood optimization.

To quantify the effect of growing pangenomes, we tested extended reference panels from 1 to 5 times larger than the initial by adding haplotypes with random mutations, and measured runtime (see "Runtime and memory usage"). As Supp. Figure 9 shows runtime grows linearly in practice, with genotyping based on the 5 times larger reference panel requiring less than 3 times longer runtime.

Comment 1.5: The comparison of Locityper with T1K could be unfair, as they are using very distinct reference sequences. It might be unfair for Locityper on HLA-A, -B, and -C genes that are well curated in the IMGT/IPD-HLA database which T1K uses. It might be unfair for T1K in other genes like HLA-J where HPRC might provide better representation than IMGT/IPD-HLA.

Since IMGT/IPD-HLA and IPD-KIR contain the genome region sequence, the authors may consider using them as the reference in Locityper to show that Locityper is competitive in HLA or KIR genotyping.

On the other hand, T1K can also take the plain genome sequence in FASTA format as the reference sequence. Though T1K is not tuned to genotype all the CMR genes with the full intronic sequences, the authors may consider testing T1K's performance using the loci genome sequences extracted by Locityper as the reference FASTA file. This comparison may demonstrate that Locityper's computational model is better for this task and address the claim "Even though these methods can achieve high accuracy, they typically rely on specific gene structure and cannot be easily scaled to include more targets." In the introduction section.

Response: We thank the reviewer for this comment. Even before the first submission we tried to set up such a comparison, however, T1K and Locityper make use of widely different input sequences. First, Locityper works well with longer sequences, which contain padding, are often tens of kilobases long, and may contain haplotypes of several nearby genes. On the other hand, T1K expects concatenated exon sequences, which are quite short, and processes multiple genes at once in order to place ambiguously mapped reads on the appropriate gene. Consequently, we were unable to run T1K with Locityper reference panels as they were probably too long—T1K spent several hours before panicking with segmentation fault. In the other direction, we were able to run Locityper with the IMGT/IPD panel, but it showed lower accuracy even compared to unweighted LOO evaluation (correct protein product: 78.8% → 57.5% for HLA-A; 64.2% → 42.0% for HLA-B; 93.8% → 38.8% for HLA-C). This is not surprising as Locityper was not designed for this use case. So while we agree with the reviewer that the evaluation we present is comparing different settings (which you might call "unfair" from a perspective of the tools), we argue that this is the most relevant comparison for users of the tools as the comparison reflects the intended way to run each tool.

Comment 1.6: Some KIR genes and HLA genes may be totally missing in a person, and Locityper correctly predicted that from the sequencing data. The description of the method does not explicitly model the case of missing a gene on both chromosomes. Is capturing gene missing through the filter using the potentially incorrect prediction flag?

Response: Genes are padded in advance (see Locus selection), sometimes several nearby genes are processed together. Consequently, Locityper can predict a missing gene by selecting a padded haplotype that does not contain the gene of interest. To clarify the issue, we added a sentence "As such, Locityper can predict missing genes by selecting padded haplotypes that lack the gene of interest."

Comment 1.7: The method section is a bit confusing. The joint likelihood is based on the equation $P(g,T|R)=P(g|T)P(T|R)$ below (3). However, $P(T|R)$ defined in (3) also depends on g through the term $L_r(g)$. The formulation of the likelihood function may need some clarification.

Response: We thank the reviewer for noticing this error. We have now replaced $P(g|T)$ with

P(CN(g) = 1 | T), and updated equations 4 and 7.

Reviewer #2 (Remarks to the Author):

This interesting work describes Locityper, a software tool designed for genotyping highly polymorphic loci in the human genome using both short- and long-read sequencing data. The authors address significant challenges in resolving genomic regions that are typically difficult to analyze due to structural complexity. Their basic idea is to optimize the alignments of *read sets* to haplotype pairs. Unlike other approaches that rely on alignments optimized for each read individually, Locityper optimizes alignments of all reads at a locus simultaneously. Results convincingly show the approach's effectiveness, significantly expanding the number of challenging medically-relevant genes available for genotyping and outperforming specialized software for HLA and KIR genes. It is striking to see that using pangenome haplotypes (enhanced new data) can lead to major genotyping improvements while the use of genome graphs (concepts for data representation) has only led to marginal genotyping improvements so far. This is in line with previous genotyping results using the T2T genome compared to older reference genomes. Overall, I found the manuscript to be of high quality, the authors address an important challenge, results are impressive, and the software tool has already been used in other projects (DOI 10.1016/j.ajhg.2024.06.007S and 10.1101/2024.04.18.590093).

Response: Thank you for this positive assessment of our work.

I would like to raise the following four major points:

Comment 2.1: The results should focus on the "leave-one-out" setting rather than the "full-database" setting, i.e., further emphasize the "leave-one-out" experiment. Locityper was exclusively tested on HPRC datasets using HPRC reference haplotypes in a "full-database" and a "leave-one-out" setting. In the "full-database" setting, the true haplotypes of the genotyped genomes are part of the reference haplotype set. This is useful for establishing a baseline in an ideal setting. In the "leave-one-out" setting, the true haplotypes are removed from the reference set. The "leave-one-out" setting mimics real application scenarios, where the true haplotype is not guaranteed to be part of the reference set. Figure 3 currently presents the (overwhelming) results for the full database, whereas the leave-one-out results (still impressive compared to the 1KGP call set) are hidden in Suppl. Fig. 2. In addition, the comparison to the 1KGP call set in the Pangenie section is unfair as both Locityper and Pangenie were run with the full database.

Response: We thank the reviewer for this suggestion. The point is well taken and Reviewer 1 made a similar note in Comment 1.1. We have now focused mainly on the LOO analysis, and moved the full-database comparison to the supplement.

To switch to a LOO setting also for the Pangenie+1kGP comparison, we would have to rerun

Pangenie in a LOO mode, which would be a time consuming step as Pangenie can only be run genome wide. As we outline also above in our response to Comment 1.1, this would in principle be doable, but the added value of this analysis would likely be low and we instead moved the corresponding paragraph to the supplement. Even though Locityper compares favorably to both Pangenie and 1kGP, we note that comparing a discovery set of variants to a regenotyped set in this variant-centric manner can be misleading due to potential differences in variant representation, which would likely be even more pronounced in a LOO setting. We think that comparing the 1kGP set to Locityper LOO in the haplotype-centric evaluation presented in Figure 3 is the correct way least affected by alignment artifacts and differences in representation.

Comment 2.2: Along the same lines, the authors use a measure they term "lost accuracy" (e.g. Fig. 2f) for evaluating the "leave-one-out" results. While the main text defines the "lost accuracy" as the "difference between best possible and predicted QVs", the Methods section on page 23 reveals a modification of this difference for "very good possible haplotypes". I found out about the modification only by digging deep into the Methods section, there is no hint to it in the main text. I strongly recommend finding another way to separate the error for "very good possible haplotypes" from the error for "less good possible haplotypes".

Response: We thank the reviewer for raising this point. We decided to remove the definition of "lost accuracy" as a derived statistic and instead show the full underlying data in Figure 2h+i. This now also visualizes a trend that when the best available haplotype is diverged (e.g. <QV25) then Locityper is more likely to lose accuracy, i.e. the space between 0 and -5 lost QV points is more occupied towards the left of the plot. This is in line with the excellent results for the "full database" case and again highlights the potential of Locityper with growing databases.

Updated Figure 2h+i:

Comment 2.3: While the evaluation of a 1KGP call set nicely establishes baseline performance of a standard pipeline for short-read data (Page 6), the benefit of using Locityper on long-read data compared to a standard pipeline remains unclear. Long read data simplifies analysis of many complex genomic loci. How does a simple standard pipeline (e.g. minimap2+sniffles) compare to Locityper on long reads?

Response: We thank the reviewer for this suggestion. We performed phased variant calling with Sniffles as well as Sniffles+Deepvariant, and included it in the Figure 2, as well as updated text in the “Locityper significantly outperforms state-of-the-art variant calling pipelines” section.

Comment 2.4: The authors advertise a fast version of Locityper requiring less than 6 minutes for HLA and KIR genotyping. This will be of tremendous interest for large sequencing projects with already aligned WGS data where computational time is expensive. Is this version similarly accurate? More generally, what is the impact of read recruitment accuracy on genotyping accuracy?

Response: We agree that this is an important point to examine. We now added CRAM-based genotyping accuracy to Supp.Table 2, and added text “Instead of unmapped reads, Locityper can process existing alignments, significantly accelerating the read recruitment stage. This does not lead to lower accuracy, as Locityper predictions for 10 mapped WGS datasets showed virtually identical results (median QV=35.25).” Note, that we also updated the corresponding runtime to 10 minutes as we now process a bigger set of MHC/KIR loci (see response to Comment 1.3).

Description	Seq.technology	Num. of samples	Total haplotypes	Mean QV	Median QV	QV ≥ 43		QV 33–43		QV ≥ 33	QV 23–33		QV 17–23		QV < 17	
						Num.	%	Num.	%	%	Num.	%	Num.	%	Num.	%
Locityper (LOO)	Illumina	40	20,350	33.60	35.27	3,102	15.24	8,865	43.56	58.81	5,491	26.98	1,853	9.11	1,039	5.11
	Simulated SR	40	22,383	34.17	35.65	3,968	17.73	9,616	42.96	60.69	5,947	26.57	1,946	8.69	906	4.05
	HiFi	20	10,173	35.24	36.90	1,901	18.69	4,870	47.87	66.56	2,427	23.86	678	6.66	297	2.92
	ONT	20	10,173	34.78	35.95	1,461	14.36	5,101	50.14	64.50	2,698	26.52	710	6.98	203	2.00
Locityper (LOO), CRAM input	Illumina	10	5,091	33.65	35.25	814	15.99	2,170	42.62	58.61	1,387	27.24	454	8.92	266	5.22

Minor points for further improving the manuscript:

Comment 2.5: While the basic idea of the method in Locityper is rather simple, its implementation involved choosing many distributions, parameters and thresholds. I am wondering how robust the choice of these variables are. What are the variables that influence the results the most apart from the Omegas?

Response: Many of the parameters control accuracy-runtime tradeoff (optimization method, number of attempts, threshold and number of discarded genotypes). Read recruitment arguments (such as match fraction) affect precision/recall of the recruitment, and can affect both runtime (if too many reads are recruited and have to be discarded) and accuracy (not enough reads are recruited). Based on preliminary experiments, WGS preprocessing parameters were generally very robust. Unmapped penalty, alignment likelihood difference threshold and window weight parameters were important for likelihood definition, and could affect accuracy at particularly complex loci. In general Locityper parameters are quite robust, work well with various sequencing technologies and at different loci, as we show in our evaluation; suboptimal parameters lead to slower runtime or slightly lower average/median accuracy, rather than significant drop in accuracy. While a comprehensive search for all parameter combinations could be done in principle, we refrained from doing this as this would be computationally heavy and unlikely to yield significant improvements.

Comment 2.6: The discussion section highlights strong and weak points of the approach (e.g., its departure from the variant-centric approach and excellent agreement on trio data as well as the limited size of the currently available haplotype panels or challenges in genotyping loci with

significant homology to other parts of the genome), thereby providing valuable guidance on what a user can expect from Locityper now and in the future. Is it possible to apply Locityper on any canonical or complex, biallelic or multiallelic variant? Would it be possible and beneficial to use it genome-wide for SV genotyping?

Response: While a genome-wide use of Locityper would in principle be possible, this is not what it was designed for and the runtime would not be very fast. An important motivation for starting our work on Locityper in the first place were the remaining limitations of Pangenie, which is a k-mer based tool for pangenome based genome inference in a genome-wide manner. Going forward we envision that a combined workflow of Pangenie and Locityper might be most beneficial in practice where all parts of the genome amenable to Pangenie-based genotyping are handled by Pangenie and especially difficult loci are targeted with Locityper. Such a workflow would be a attractive tradeoff between accuracy and scalability.

Comment 2.7: Page 3 "We show that finding a maximum likelihood read assignment can be formulated as an integer linear programming (ILP) problem (Methods), for which Locityper employs existing ILP solvers and stochastic optimization."

This sentence is misleading. In my understanding, Locityper does not employ stochastic optimization for solving the ILP but it either solves an ILP or employs stochastic optimization for finding a maximum likelihood read assignment.

Response: We thank the reviewer for this comment and we changed the text to "We show that finding a maximum likelihood read assignment can be formulated as an integer linear programming (ILP) problem or identified through stochastic optimization"

Comment 2.8: Page 4 "For each target locus we used a reference panel of up to 90 haplotypes"

What does "up to" mean? Please provide mean and variance of number of haplotypes that remain after preprocessing target loci. Does the number of haplotypes correlate with genotyping accuracy?

Response: Number of haplotypes can be smaller for two reasons: there are identical haplotypes, or there are not fully-assembled haplotypes. We added the number of haplotypes to Supp. Table 1, on average there are 80.8 haplotypes per locus. There is indeed a significant negative correlation (Spearman's $\rho = -0.59$, $p\text{-value} < 2.2 \cdot 10^{-16}$) between average QV (HiFi, LOO) and the number of haplotypes.

Comment 2.9: Page 5 "consequently, we will split genotypes into those that failed and those that passed filtering."

Please reference Methods section.

Response: We have now completely removed post-genotyping filtering as it resulted in a more complex manuscript without generating significant accuracy improvement.

Comment 2.10: Page 5 "out of a total of 20,350 fully assembled sample-locus haplotypes across the 256 CMR loci and 40 Illumina WGS samples"

Why 20,350 in total? $256 \times 2 \times 40 = 20480 \neq 20350$

Response: Comparison was only performed on fully assembled haplotypes.

Comment 2.11: Page 7 "In addition to the HPRC dataset, we genotyped the full 1KGP cohort of 3202 Illumina WGS samples, including 602 trios"

Does this cohort include the 39 HPRC samples that is mentioned to overlap the 1KGP cohort on page 6? If yes, do you see better concordance for these samples, where the haplotype is part of the reference haplotypes?

Response: 1KGP trios indeed contain HPRC samples; we have now removed them from the concordance analysis and process 563 trios. This only slightly affects overall concordance (median QV 44.64 \rightarrow 44.43). At the remaining 39 trios, Locityper has indeed achieved a higher median QV of 47.12.

Comment 2.12: Page 12 "Locityper employs integer linear programming and stochastic optimization"

Should read "Locityper employs integer linear programming or stochastic optimization".

Response: We thank the reviewer for this comment and changed the sentence as suggested.

Comment 2.13: Page 17 "A read pair is retained if both read ends have at least one good alignment ($p \geq 0.01$) to at least one of the haplotypes. All alignments with BB $p < 0.001$ are discarded" and Page 20

How many alignments per read are retained per haplotype pair on average? How big is the search space for optimizing read assignment? Does this number somehow influence the predicted genotype's accuracy?

Response: We examined the fraction of retained read pairs per locus for HG00438 Illumina dataset, and in most loci only a minor fraction of reads was discarded (median 3.3%, 90th-percentile = 6.2%). *SLC29A4*, *DPY19L2*, *ESPN*, *KMT2C* were four loci with the highest fraction of discarded reads (20-30%), likely due to reads incorrectly recruited from homologous regions. The fraction of discarded reads only slightly correlates with LOO-QV (Spearman's $\rho = -0.24$). Search space for read assignment varies depending on the locus and input sample, but in general each read pair has at least three possible locations for close-to-optimal genotypes (≥ 1 location on each haplotype + unmapped location). For the HG00438 Illumina dataset, on average 5.5 thousands read pairs were used for locus genotyping, resulting in approximately 3^{5500} possible read assignments for candidate genotypes.

Comment 2.14: Page 17 "For null windows, we define insert size probability as the highest probability achievable under the precomputed insert size distribution."

Why the highest?

Response: We added a sentence “This way, insert size between a read end and its unmapped counterpart is assumed to be optimal in order to only penalize unpaired locations once.” In practice, the insert size distribution is quite wide and most read pairs have insert probabilities, which are close to the highest. Therefore it does not make sense to penalize unpaired locations even more, especially since it is hard to define a fair insert size probability penalty and in our view does not make sense to add another parameter just for this case.

Comment 2.15: Page 17 " Finally, we will denote the full set of possible read pair locations on haplotype h as $L(h)_r \subset W(h) \times W(h)$ "

Why do you use L_r instead of just L ? Did I miss that L somehow depends on r ?

Response: We thank the reviewer for this comment, we have updated the notation and removed “ $_r$ ”.

Comment 2.16: Page 19 "The two values need to be defined in advance and should sum up to 2"

Why do they need to sum up to 2? Don't they scale the likelihoods for all haplotype pairs in the same way so that you will obtain the same results even if they don't sum up to 2?

Response: In terms of pure optimization, the two values can have any positive values. However, in the context of likelihoods, it makes sense to keep the same range of values. While these values are barely used, they can still be helpful in rare edge cases (for example if less than 3 optimization attempts are used per genotype and no likelihood variance is available).

Comment 2.17: Page 21 "weighted by the corresponding genotype probabilities"

How do you compute genotype probabilities? Do you normalize the genotype likelihoods?

Response: We added a sentence “Additionally, we redefine genotype probabilities as the probability of having the highest true likelihood, calculated as the product of inverse t-test p-values for all pairwise genotype comparisons”. This is a heuristic approximation that may be changed in the future, and predicted probabilities are no longer used in the paper.

Comment 2.18: Page 21 "we marked genotypes as potentially incorrect if weighted distance is over 30"

Can you use a fixed threshold across all loci? Doesn't this depend on the similarity of the haplotypes in the reference set?

Response: Our initial idea was that this would allow us to identify ambiguous genotype predictions, where several distinct genotypes are predicted with high probability. Therefore, overall divergence of the reference set does not matter: if all haplotypes are similar to each other we cannot use this metric; and if all haplotypes are very distinct—it should be easy to identify ambiguous predictions. However, we have since removed this section from the manuscript (see response 2.9).

Comment 2.19: Supp. Table 1

Please indicate the reference genome for the given coordinates.

Response: We have updated Supp. Table 1.

Reviewer #1 (Remarks to the Author):

The authors have addressed my concerns. Happy to see the review is helpful for the locityper wegithed mode.

A minor comment about “T1K often predicted smaller copy number than required” in the HLA/KIR benchmark. This is because T1K by default does not report copy number variations, so homozygous HLA/KIR alleles will only report as one. If copy number is needed, users may use “t1k-copynumber.py” in the T1K package to infer the copy number. This script is described in the “immuannot”’s publication, which the author also used in the evaluations. The authors have done a nice comparison in Figure S8 that can handle the copy number issue, so this comment is minor and just to provide some background information.

We thank the reviewer for this information. We will compare against T1K copy number-modified script in our future benchmarks.

Reviewer #2 (Remarks to the Author):

The authors responded to all issues raised by me and the other reviewer and appropriately addressed all my major concerns by focussing on the more realistic LOO experiment and adding further assessments showing Locityper's superior performance. Again, I found the manuscript to be of high quality and the results are excellent. All remaining points are minor.

While the authors *replied* to all my initial minor comments, comments 2.5, 2.6, 2.8, 2.13 and 2.16 have not been addressed by making changes to the manuscript. This is not critical but unfortunate as the responses contain insights that I consider interesting for readers.

To address comment 2.5 we added two sentences to the main text:

“When needed, users can provide custom Ω values to adjust read alignment/depth balance for specific loci of interest in order to achieve optimal accuracy.”

“Lower match fraction values lead to an increased number of unnecessarily recruited reads, which increases read mapping runtime, but does not significantly affect genotyping accuracy since false positive reads are discarded at a later stage.”

To address comment 2.6 we now added a paragraph “Over the full genome, Pangenie shows very high genotyping accuracy; however, its reliance on unique k -mers results in lower calling power at especially challenging loci. We envision that genome-wide Pangenie variant calling can be complemented by targeted Locityper analysis to produce scalable and accurate genome-wide variant calling workflow.” to the Supplementary Information.

For 2.8 we added: “Even though reference panels were constructed based on 90 haplotypes from whole genome phased assemblies, on average around 80 unique haplotypes were reconstructed per locus, as some haplotypes are not unique, while others are only partially

\ n + ž ž (.

e (. (. n (ž (ž

i ž (. (((ž +

. (ž (. i > j n b n : ž

\ ž ž n ž ž b b b D ž ž n n n ž ž ž

Г ž ž (. * ž ж b ž : (. \ j # (. ž b ž

\ + (. n (. n (.

2 1 (2 ž ž +: (. (ž () ((. (. + : (. (ž + (. ((.

() ž (. n \ (. * # n +: (. (ž (.

\ (. +: (. (a (n + (. (ž ((. + (. (. ž

: (. n : ž ž (. \ j # (. (. (.) n +

2 b ž (. n) (. + e (. (. 2 + + ((ž \ 1 (. + (.

(. + (. (. (.) +: (. ((.